# A computational approach to map nucleosome positions and alternative chromatin states with base pair resolution

Xu Zhou[1,2,3†‡], Alexander W Blocker[4†], Edoardo M Airoldi[4,5*], Erin K O'Shea[1,2,3,6*]

[1]Department of Molecular and Cellular Biology, Harvard University, Cambridge, United States; [2]Faculty of Arts and Sciences Center for Systems Biology, Harvard University, Cambridge, USA; [3]Howard Hughes Medical Institute, Harvard University, Cambridge, United States; [4]Department of Statistics, Harvard University, Cambridge, United States; [5]The Broad Institute of MIT and Harvard, Cambridge, United States; [6]Department of Chemistry and Chemical Biology, Harvard University, Cambridge, United States

**Abstract** Understanding chromatin function requires knowing the precise location of nucleosomes. MNase-seq methods have been widely applied to characterize nucleosome organization in vivo, but generally lack the accuracy to determine the precise nucleosome positions. Here we develop a computational approach leveraging digestion variability to determine nucleosome positions at a base-pair resolution from MNase-seq data. We generate a variability template as a simple error model for how MNase digestion affects the mapping of individual nucleosomes. Applied to both yeast and human cells, this analysis reveals that alternatively positioned nucleosomes are prevalent and create significant heterogeneity in a cell population. We show that the periodic occurrences of dinucleotide sequences relative to nucleosome dyads can be directly determined from genome-wide nucleosome positions from MNase-seq. Alternatively positioned nucleosomes near transcription start sites likely represent different states of promoter nucleosomes during transcription initiation. Our method can be applied to map nucleosome positions in diverse organisms at base-pair resolution.

*For correspondence: airoldi@fas.harvard.edu (EMA); osheae@hhmi.org (EKO)

†These authors contributed equally to this work

Present address: ‡Yale School of Medicine, New Haven, United States

## Introduction

The eukaryotic genome is compacted into chromatin (*Kornberg, 1974*) which is comprised of nucleosomes, each consisting of approximately 147 base pairs (bp) of DNA wound around a histone protein octamer (*Kornberg and Lorch, 1999*). The helical DNA makes direct contact with the histones every 10 base pairs, with the major groove of DNA alternating between facing towards and away from the histone core (*Luger et al., 1997*). Shifting the histones relative to the DNA sequence by a few base pairs can change the accessibility of sequence elements to DNA binding proteins if they are located in the linker sequences between nucleosomes, or may switch these elements between facing towards and away from nucleosomes if they are located within nucleosomal DNA (*Jiang and Pugh, 2009b*; *Segal and Widom, 2009b*; *Zhang and Pugh, 2011*).

The location of nucleosomes with respect to DNA sequences influences many biological processes. Nucleosomes restrict the accessibility of DNA sequences to protein factors, such as transcriptional regulators and the transcription machinery (*John et al., 2011*; *Li et al., 2007*; *Liu et al., 2006*; *Zhou and O'Shea, 2011*). The positions and occupancy of nucleosomes can influence the interplay between transcription factors (*Mirny, 2010*) and the level (*Carey et al., 2013*; *Kim and O'Shea, 2008*), dynamics (*Lam et al., 2008*), and differences in gene expression between cells

**eLife digest** Plants, animals and other eukaryotes wrap their DNA around complexes of proteins called histones to form repeating units known as nucleosomes. The interaction between histones and DNA is strong, and so the DNA region inside a nucleosome has limited access to other proteins, including those that drive the expression of genes.

Moving a nucleosome slightly can change the access to its DNA and significantly impact how the genes in the region are regulated. Nevertheless, determining the position of nucleosomes accurately or testing how nucleosomes are different between individual cells are challenging tasks. Most methods for identifying nucleosomes use an enzyme called micrococcal nuclease (or MNase for short) to break down the DNA that isn't protected in nucleosomes, followed by high-throughput DNA sequencing to identify the DNA fragments that remain. However, this technique, known as MNase-seq, is limited because it only measures an average location of the nucleosomes across millions of cells.

Now, Zhou, Blocker et al. have developed a new computational approach to identify nucleosome positions more accurately using MNase-seq data obtained from both yeast and human cells. This approach revealed that in more than half of the yeast genome, a given nucleosome is found at slightly different positions in different cells. Nucleosomes positioned near the beginning of a gene mark it open or closed for binding by the cell's gene expression machinery. Zhou, Blocker et al. suggest that the nucleosomes' positions influence how gene expression starts via a multi-step process.

Following on from this work, the next step is to use the newly developed method to study how nucleosome positions change when other regulators of gene activity bind and when genes are activated or repressed.

(*Dadiani et al., 2013*; *Raser and O'Shea, 2004*; *Tirosh and Barkai, 2008*). Recently, nucleosome organization has also been suggested to affect how gene promoters interpret dynamic signaling information at the single cell level (*Hansen and O'Shea, 2013*; *Hao and O'Shea, 2012*), and heterogeneity in promoter nucleosome positions has been linked to differences in gene expression (*Small et al., 2014*). Knowing the precise location that nucleosomes occupy with respect to DNA sequence is crucial for understanding how these biological processes are influenced by eukaryotic chromatin.

Genome-wide nucleosome positions are commonly mapped with micrococcal nuclease digestion based high-throughput sequencing (MNase-seq) (*Hughes and Rando, 2014*). In this method, histone-DNA interactions protect DNA from MNase digestion and the protected DNA fragments are sequenced and aligned to genome sequences to infer the location of nucleosomes (*Clark, 2010*; *Kaplan et al., 2009*; *Rando, 2010*; *Zhang and Pugh, 2011*). Although MNase exhibits sequence preference when digesting DNA devoid of histones (*Horz and Altenburger, 1981*), genome-wide analyses of nucleosomes with MNase-based methods are generally consistent with studies using MNase-independent methods (*Hughes and Rando, 2014*), such as DNase I chromatin digestion (*Hesselberth et al., 2009*) and chemical cleavage (*Brogaard et al., 2012*).

Studies that apply MNase-based methods typically report the position of a nucleosome as the average of the bulk nucleosome population (referred to as the 'consensus center', *Figure 1A*) (*Struhl and Segal, 2013*; *Zhang and Pugh, 2011*). However, if nucleosomes have overlapping positions in a significant portion of the population, the effect of averaging over heterogeneous nucleosome positions can lead to discrepancy between the consensus center of nucleosomes and the most representative nucleosome positions (*Figure 1A*). A variety of methods have been developed to improve the precision of nucleosome mapping from MNase-seq data, such as peak finding of nucleosome occupancy (*Zhang and Pugh, 2011*) and filtering of single-end digestion patterns (*Weiner et al., 2010*), but determining the precise locations of individual nucleosomes within a cell population remains a challenge due to substantial variability in the mapped locations of digested nucleosomes – the midpoints of paired-end sequenced nucleosomes or the endpoints of single-end sequenced nucleosomes (*Figure 1B*). This variability may arise from a cluster of overlapping and

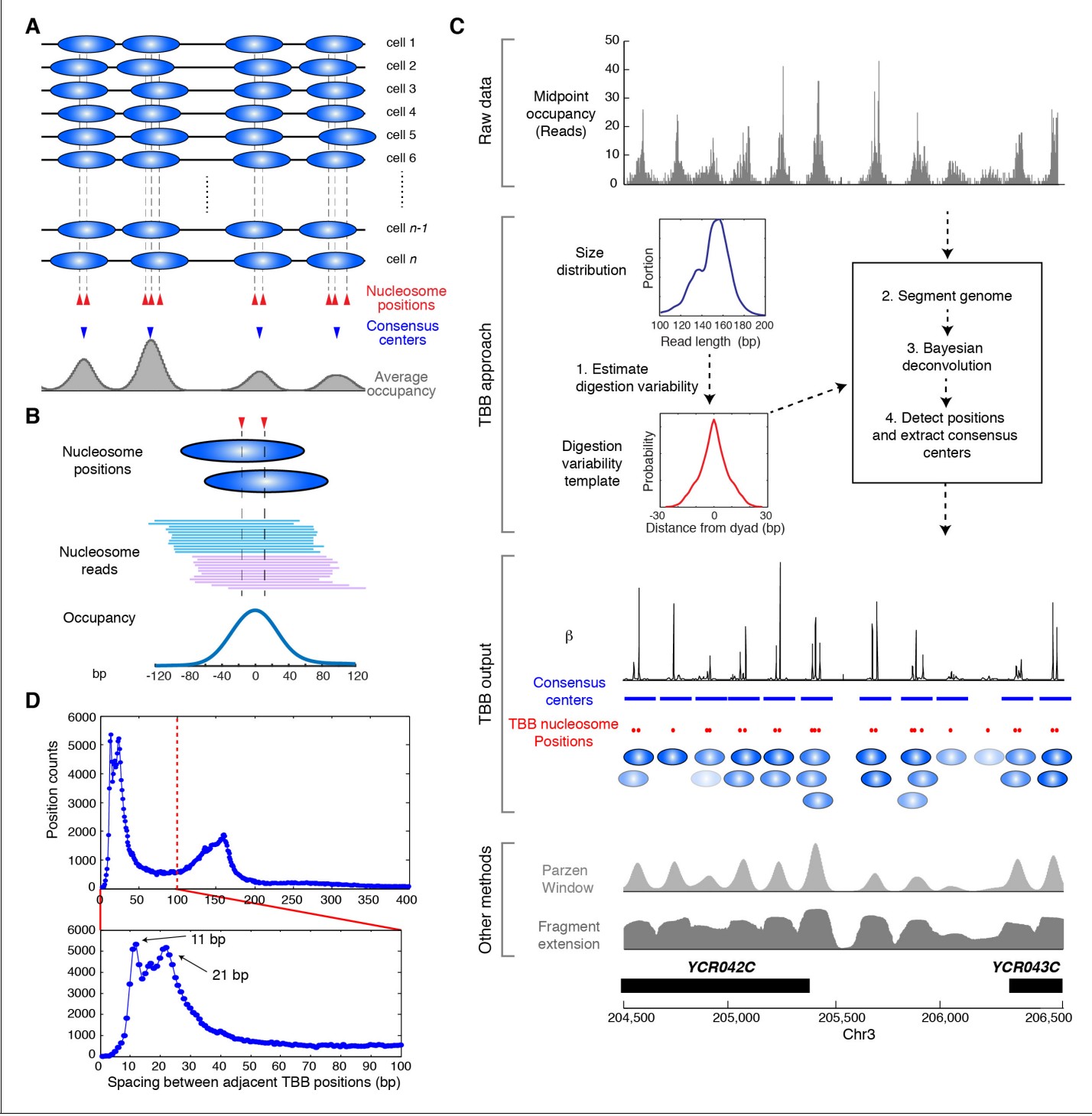

**Figure 1.** Illustration of the Template-Based Bayesian (TBB) approach for determining nucleosome positions. (**A**) Diagram illustrating the heterogeneous nucleosome positions and the consensus centers of nucleosomes along a genomic region in a population of cells. Blue ovals illustrate individual nucleosomes and dotted lines mark all nucleosome positions. (**B**) Example of digested nucleosome reads, their nucleosome positions and the overall occupancy. (**C**) Illustration of the computational pipeline of the TBB approach. Occupancy of sequencing read midpoints indicates the number of midpoints at every base pair for yeast Chr 8, 204, 500–206,500 bp. Blue ovals illustrate overlapping TBB nucleosome positions and are colored according to the magnitude of their coefficients β. Two common presentations of nucleosome sequencing data are shown for comparison: the light gray area represents the nucleosome occupancy generated by smoothing sequencing read midpoints with a Parzen window approach (band size of 20 bp) (*Albert et al., 2007*; *Tsankov et al., 2010*); the dark gray area (Fragment extension) represents the nucleosome occupancy generated by

*Figure 1 continued on next page*

*Figure 1 continued*

extending 73 bp on both ends from the sequencing read midpoints. (D) Histogram showing the distance between adjacent TBB nucleosome positions in a combination of the T1 and T2 experiments.

The following figure supplements are available for figure 1:

**Figure supplement 1.** Diagrams of nucleosome digestion variability template estimation.

**Figure supplement 2.** Length distribution of nucleosome reads.

stably positioned nucleosomes, as well as from effects causing different degrees of digestion of the same nucleosome by MNase, such as nucleosome breathing and nuclease trimming – all of which influence the distribution of the aligned reads and are difficult to disentangle (*Clark, 2010*). Recently, a chemical cleavage approach that uses a genetically engineered histone H4 to chemically cleave DNA sequences in contact with the nucleosome dyad allowed direct measurement of nucleosome positions with unprecedented resolution (*Brogaard et al., 2012*; *Moyle-Heyrman et al., 2013*). However, the requirement for genetic engineering of essential histones limits its current application to genetically tractable organisms. Therefore, novel experimental or analytical approaches that are generally applicable in eukaryotes are still needed to determine the accurate positions of nucleosomes in vivo. Here, we report a computational approach to determine in vivo nucleosome positions from paired-end MNase-sequencing data.

Applying template-based deconvolution to experimental data has many applications in biology. For example, in super-resolution fluorescence microscopy, the locations of individual fluorophores within the diffraction limit of light can be identified by deconvoluting light intensity information with a function describing the distribution of light intensity from individual light spots (*Betzig et al., 2006*; *Huang et al., 2009*; *Rust et al., 2006*). Inspired by this, we use the size distribution of MNase digested nucleosome fragments to infer a digestion variability template for nucleosomes, and report a Bayesian method that makes use of these templates to identify the individual positions of nucleosomes at a base-pair resolution, hereafter referred to as the template-based Bayesian (TBB) approach. This approach can be applied to data generated through both paired-end and single-end sequencing to map chromatin structure in diverse organisms. Here we demonstrate the template-based Bayesian approach with paired-end sequencing of MNase digested nucleosomes in yeast and human cells. We show that the periodic occurrences of dinucleotide sequence motifs relative to the nucleosome dyad can be directly determined from MNase based nucleosome positions and are conserved in vivo in both yeast and human cells. Leveraging this method, we find that alternative configurations of nucleosomes are a common feature in both yeast and human chromatin. The alternatively positioned nucleosomes around gene transcription start sites represent configurations that differ in their compatibility with the assembly of the pre-initiation complex. A 3-step model for transcription initiation can reconcile the competition between nucleosomes and the transcriptional machinery observed from genomic analysis.

## Results

### A Bayesian approach based on digestion variability templates can identify positions of nucleosomes at base-pair resolution

We reasoned that it might be possible to resolve the positions of multiple overlapping nucleosomes if we could estimate the degree to which MNase digestion contributes to the deviation of the midpoints from a true nucleosome center. Since the overall digestion of nucleosomes is reflected in the length of nucleosomal DNA fragments, we tested the idea that we might be able to estimate the variation in the midpoints from the length of digested nucleosomes (*Figure 1—figure supplement 1*), and use this information to infer the positions of individual nucleosomes through deconvolution. The variability of digested nucleosomes could come from two sources: the technical variation that is associated with nuclease cleavage, such as variable trimming at nucleosome ends, and biological variation that directly influences the length of DNA wound around histones, such as nucleosome

breathing and remodeling (*Polach and Widom, 1995*). While the technical effects are likely to affect both ends of nucleosomal DNA equally, the biological effects may create bias for specific nucleosomes and/or a specific side of the nucleosomes. However, the biological effects are generally believed to be transient and rare at a given genomic location within a population of cells (*Andrews and Luger, 2011*); we thus assumed that the digestion variation was equivalent at both ends of the nucleosome when averaged over the genome and population. The biological variation at individual nucleosomes could generate large shifts in read midpoints due to length differences in nucleosomes (likely by multiples of 10 bp due to the unwrapping of each helical turn of DNA), and could be identified as alternative nucleosome positions if they were present in a significant fraction of the bulk nucleosome population. Nucleosomes with substantially smaller size, such as sub-nucleosomes (*Rhee et al., 2014*), can be identified based on the sequenced fragment size (<100 bp) and were excluded from our analysis.

The template-based Bayesian approach includes four major steps (*Figure 1C*, see Materials and methods). First, we estimated the distribution of MNase digestion variability from the length distribution of DNA sequencing reads, assuming that MNase digestion is not biased towards either end of the nucleosome on average. This produced a digestion variability template, describing the overall deviations in read midpoints relative to the true nucleosome center (*Figure 1—figure supplement 1*). Second, each chromosome was segmented into regions with similar sequencing coverage to increase detection sensitivity, with a requirement to maintain sufficient segment length for estimating statistical properties of nucleosome positions. Third, within each segment, the occupancy of read midpoints (the number of paired-end read midpoints at each base pair, *Figure 1C*, gray trace) was deconvoluted with the digestion variability template to estimate the expected read occupancy at every base pair if digestion errors had not occurred (coefficient β, *Figure 1C*, black trace). Finally, the locations of nucleosomes were summarized based on this expected read occupancy and statistical thresholds. To control for falsely identified nucleosome positions arising from the sequence preference of MNase, we set the statistical thresholds using a false discovery rate (FDR≤0.05) estimated based on a simulated MNase digestion dataset that matches the experimentally observed digestion sequence distribution. In our study, the length distribution of paired-end sequencing reads from the gene coding and promoter regions was indistinguishable from the rest of the genome (*Figure 1—figure supplement 1–2*). Therefore, a single digestion variability template was applied across the yeast genome.

We represented the locations of nucleosomes using two metrics: the consensus centers of nucleosomes and the TBB positions of nucleosomes (*Supplementary file 1*). The consensus centers of nucleosomes were defined to represent non-overlapping nucleosomes with a minimum size of 147 bp (*Figure 1C*, blue bars), by applying a Gaussian window smoothing method to the expected read occupancy (coefficient β) from our model – they are equivalent to the nucleosome positions commonly described in the literature (*Albert et al., 2008*; *Jiang and Pugh, 2009b*; *Struhl and Segal, 2013*) (hereafter termed 'consensus centers'). The TBB nucleosome positions describe the center locations of individual nucleosomes across the genome without limitation on minimum spacing (*Figure 1C*, red dots and blue ovals) (hereafter termed 'TBB nucleosome positions'), which represent individual stably positioned nucleosomes over a statistical threshold.

## Accurate estimation of nucleosome positions at base-pair resolution

We titrated MNase to digest yeast chromatin and selected for analysis two nucleosome samples that were digested to different sizes ('T1' and 'T2', *Figure 2—figure supplement 1*). Both samples were digested to primarily mono-nucleosomes without generating over-represented sub-nucleosome fragments. We generated paired-end data sets of 'T1' and 'T2', each with a coverage of ~2.8 read midpoints per base pair; as an average between these two data sets, we identified ~61,800 consensus centers of nucleosomes and ~125,000 TBB nucleosome positions, covering at least 75.5% of the *S. cerevisiae* genome. The spacing between adjacent TBB nucleosome positions peaked at 11 and 21 bp (*Figure 1D*), consistent with the rotation of DNA sequence by one or two helical turns around histone cores (*Albert et al., 2007*; *Brogaard et al., 2012*), and suggesting that the same nucleosome commonly occupies overlapping configurations in different cells.

The consensus centers of nucleosomes identified by the template-based Bayesian approach were consistent with the published methods employing Gaussian smoothing algorithms (often referred to as 'Parzen window') (*Albert et al., 2007*, *2008*; *Clark, 2010*; *Tsankov et al., 2010*) (*Figure 2A*),

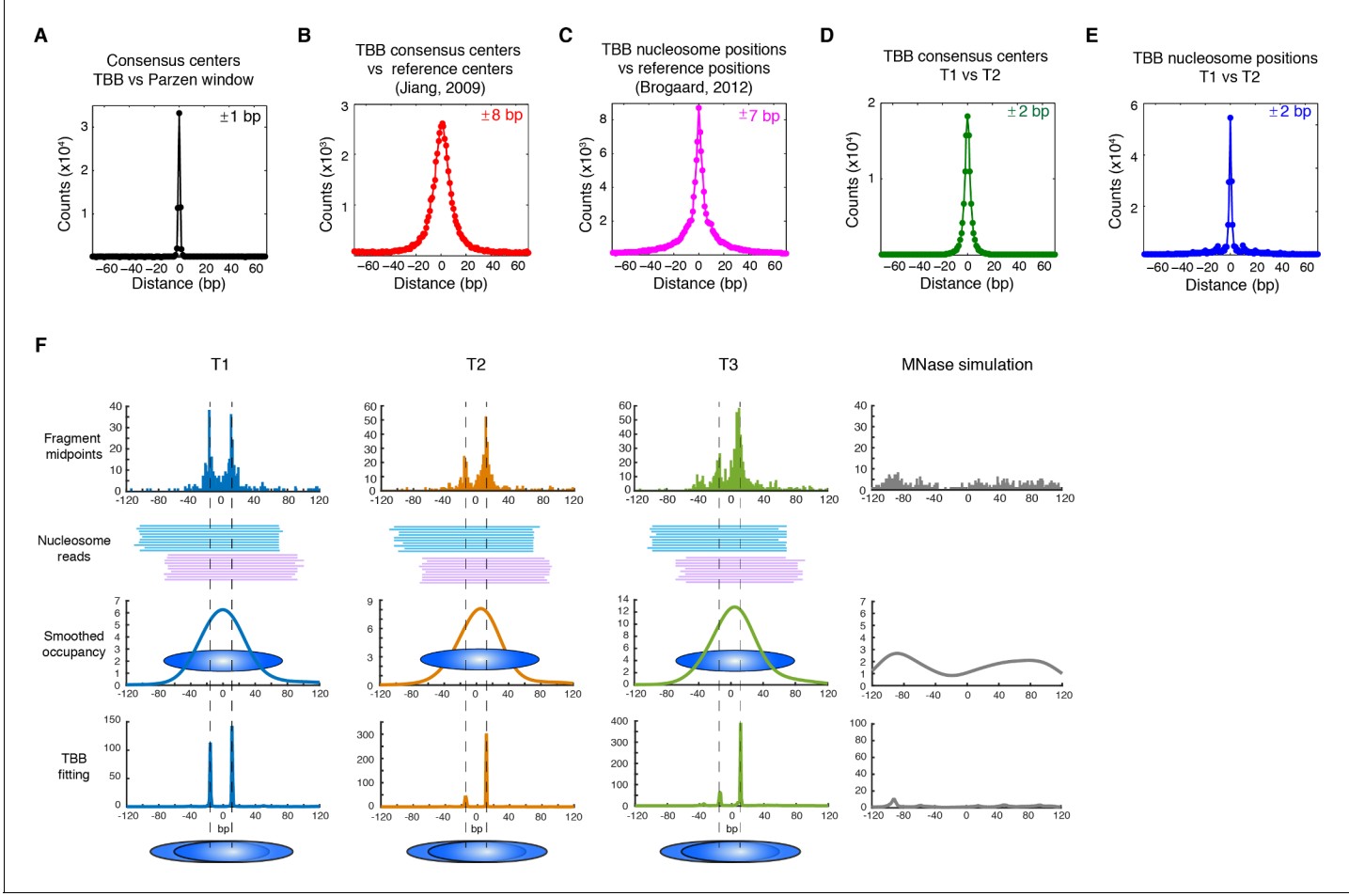

**Figure 2.** Genome-wide evaluation of the TBB approach and alternative nucleosome positions. Histogram of the nearest distance: (A) between the consensus centers of nucleosomes determined by the TBB approach and by the Parzen window approach; (B) between the consensus centers determined by the TBB approach and the reference MNase nucleosome positions (*Jiang and Pugh, 2009a*); (C) between the TBB nucleosome positions and the nucleosome positions mapped by the chemical approach (*Brogaard et al., 2012*); (D,E) between the consensus centers of nucleosomes (D) and the TBB nucleosome positions (E) mapped in two independent experiments. The median of the distance between matched consensus centers or TBB nucleosome positions is reported for each comparison. (F) Example of stable TBB positions that are tolerant to MNase digestion, chr 8, 341651 – 341771. Randomly selected reads fragments are shown to represent the locations of sequenced tags.

The following figure supplements are available for figure 2:

**Figure supplement 1.** Titration of MNase digestion.

**Figure supplement 2.** Cumulative distribution of the nearest distance analysis in *Figure 2A*.

**Figure supplement 3.** Cumulative distribution of the nearest distance analysis in *Figure 2B*.

**Figure supplement 4.** Cumulative distribution of the nearest distance analysis in *Figure 2C*.

**Figure supplement 5.** Cumulative distribution of the nearest distance analysis in *Figure 2D*.

**Figure supplement 6.** Cumulative distribution of the nearest distance analysis in *Figure 2E*.

with more than 90% of the nucleosomes matching within a single base pair (*Figure 2—figure supplement 2*). These consensus centers also showed general agreement with the published nucleosome reference maps (*Jiang and Pugh, 2009a*) (*Figure 2B* and *Figure 2—figure supplement 3*), validating the overall biological relevance of our data sets. We then evaluated the accuracy of our identified TBB nucleosome positions by comparing them to individual nucleosome positions mapped through a recently reported chemical approach (*Brogaard et al., 2012*), which provides a reference that is independent of MNase cleavage specificity. The nucleosome positions identified by these two approaches were consistent (*Figure 2C* and *Figure 2—figure supplement 4*), demonstrating that the TBB approach can precisely map the positions of individual nucleosomes. Further, we assessed the absolute precision of the TBB approach by using it to map nucleosome positions from in silico generated MNase-seq data sets where the simulated nucleosome positions are known. We systematically tested the precision of the TBB approach with varying sequencing coverages, relative nucleosome population abundance and spacing among overlapping nucleosomes (*Figure 3* and Material and Methods). The majority of the TBB nucleosome positions identified in these simulated data sets were within 2–3 bp of the originally simulated nucleosome positions, even for the cases where nucleosome positions were separated by only 10–15 bp or differ by more than 3-fold in their relative occupancy (*Supplementary file 2*).

The accuracy and reproducibility of MNase-based mapping of nucleosomes may be limited by MNase digestion variability. The degree to which chromatin is digested by MNase depends on the DNA sequence and the preparation of nucleosomal DNA samples (*Dingwall et al., 1981*; *Horz and Altenburger, 1981*), and may differ significantly between experiments even when experimental protocols have been carefully followed. To evaluate the tolerance of the TBB nucleosome positions to different levels of MNase digestion, we compared the consensus centers and TBB nucleosome positions in the 'T1' and 'T2' data sets that represent digestion to different fragment sizes (*Figure 2—figure supplement 1*). Both the consensus centers and the TBB nucleosome positions were in agreement between these two experiments (*Figure 2D–E*, *Figure 2—figure supplements 5–6*), with a median difference of 2 bp across the genome; in comparison, randomly generated consensus centers and nucleosome positions gave rise to a median difference of 49 bp and 36 bp, respectively. Individual nucleosome positions can be reliably detected regardless of the differences in MNase digestion (*Figure 2F*), demonstrating the robustness of the determined nucleosome positions to digestion variability. Overall, these results demonstrate that the positions of nucleosomes can be reliably and accurately detected at base-pair resolution with paired-end MNase-seq coupled with the template-based Bayesian approach.

Several methods have been developed to determine the genome-wide locations of nucleosomes, each of which addresses different experimental and computational issues (*Albert et al., 2007a*; *Brogaard et al., 2012*; *Polishko et al., 2012*; *Schep et al., 2015*; *Tirosh, 2012*; *Tsankov et al., 2010*; *Valouev et al., 2011*; *Weiner et al., 2010*; *Zhong et al., 2016*). We briefly summarized the approaches, performance and limitations of these methods for reference (*Supplementary file 3*).

## Dinucleotide periodicity is a conserved determinant of in vivo nucleosome positioning

Nucleosomes in vivo are thought to be enriched for genome sequences that show a pattern of periodic AA/AT/TA/TT or CC/CG/GC/GG motifs that oscillate every 10 bp with opposite phases (*Albert et al., 2007a*; *Brogaard et al., 2012*; *Gaffney et al., 2012*; *Kaplan et al., 2009*; *Mavrich et al., 2008*; *Satchwell et al., 1986*; *Segal et al., 2006*). This dinucleotide sequence pattern matches the bending property of preferred nucleosomal DNA sequences, as the double stranded DNA is twisted and bent to maintain contact with the histone cores (*Drew and Travers, 1985*). However, this periodic pattern was not observed around the consensus centers of nucleosomes determined using MNase based mapping methods (*Valouev et al., 2011*) (*Figure 4—figure supplement 1*) and only emerged from nucleosome fragments selected to be exactly 147 bp (*Figure 4—figure supplement 2*) (*Albert et al., 2007a*; *Gaffney et al., 2012*; *Kaplan et al., 2009*), perhaps due to the imprecision of the determined nucleosome centers.

In contrast, when we aligned DNA sequences at the TBB nucleosome positions in yeast, the dinucleotide AA/AT/TA/TT and CC/CG/GC/GG motifs displayed 10 bp periodicity across almost the full length of the nucleosome core (*Figure 4A* and *Figure 4—figure supplement 3*), illustrating the power of the TBB approach to precisely map nucleosome positions from MNase-based methods. To

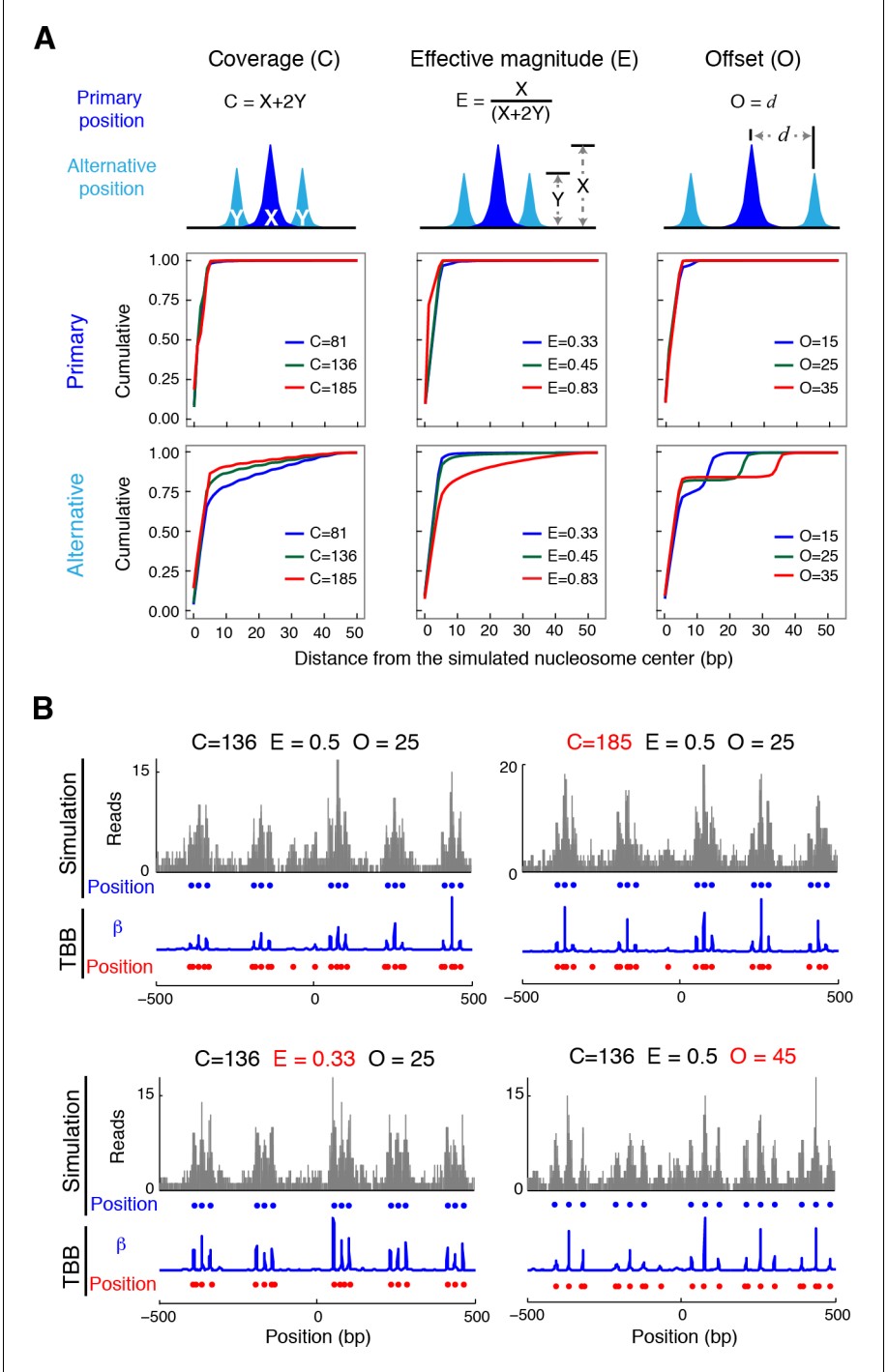

**Figure 3.** Nucleosome detection from in silico MNase-seq datasets. (**A**) Plots summarizing the distance between the detected TBB nucleosome positions in the in silico datasets and the nearest simulated primary and alternative nucleosome positions. ('**C**', total sequencing coverage of all overlapping nucleosomes; '**E**', the effective magnitude (relative occupancy of neighboring nucleosomes); and '**O**', the offset (spacing between nearby nucleosome positions). (**B**) Examples of nucleosome detection in the simulation at different coverage, effective magnitude and offset (different values are highlighted in red). Sequencing read midpoints (gray) were distributed randomly around the simulated nucleosome positions (blue dots) according to the digestion variability template. The coefficients (blue trace) and nucleosome positions (red dots) determined by the TBB approach are shown for comparison.

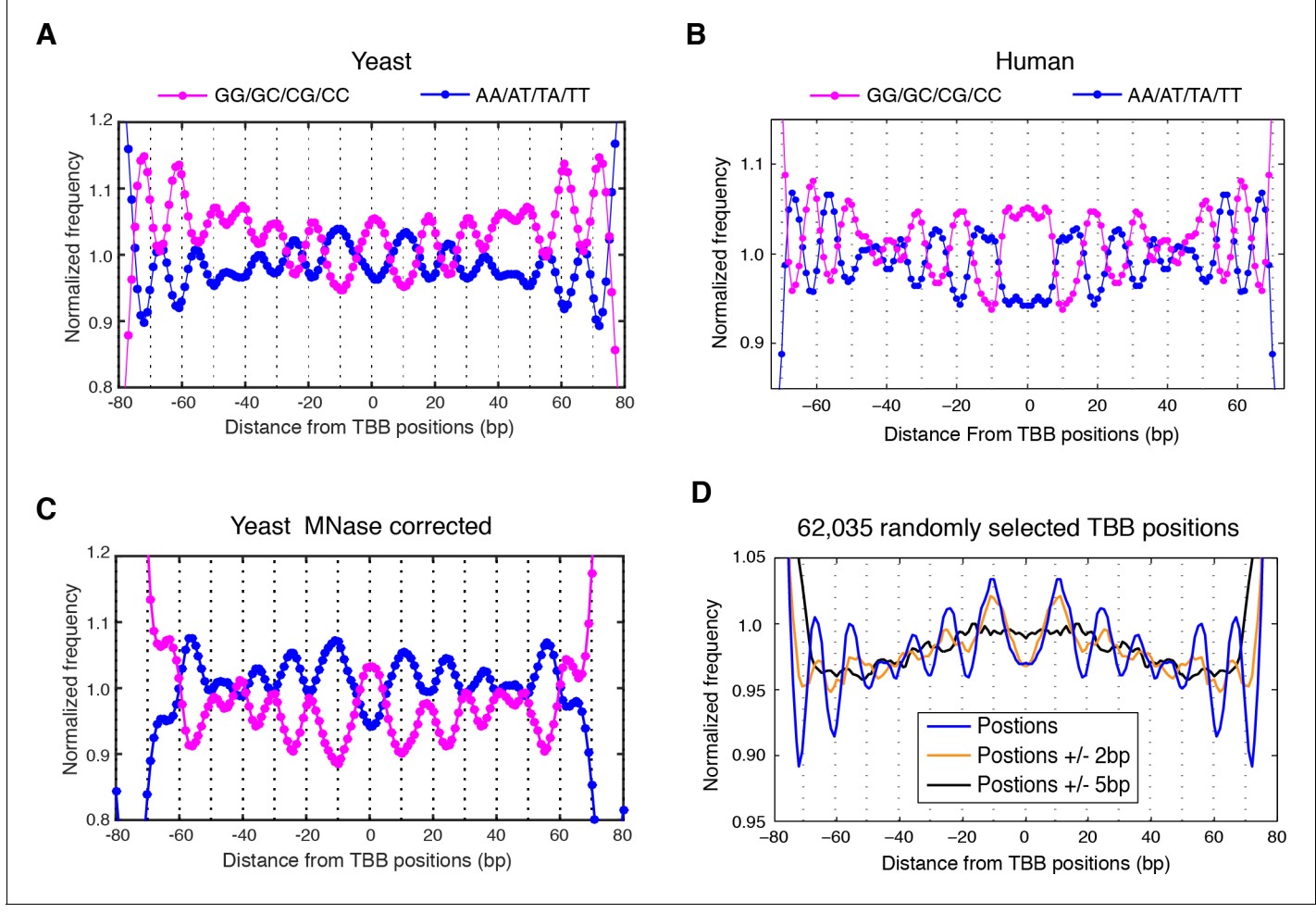

**Figure 4.** Dinucleotides frequency of nucleosome positions. (A–C) Normalized frequency of AA/AT/TA/TT and CC/CG/GC/GG dinucleotides of DNA sequences aligned at the centers of nucleosomes, for all TBB nucleosome positions in yeast, both before (A) and after (C) correction for MNase digestion bias, and for all TBB nucleosome positions on human chromosome 12, position 38,000,000 bp to 48,000,000 bp, a 10 Mbp region randomly chosen in the human genome (B). (D) The frequency of AA/AT/TA/TT for 62,035 randomly selected TBB nucleosome positions (blue trace), and for genome locations with an average distance of either 2 bp (orange trace) or 5 bp (black trace) from these selected TBB nucleosome positions. The distance was randomly perturbed by between 0–4 bp or 0–10 bp for each nucleosome positions, respectively.

The following figure supplements are available for figure 4:

**Figure supplement 1.** Dinucleotides frequency of nucleosome consensus centers.

**Figure supplement 2.** Dinucleotides frequency of selected 147 bp nucleosome reads.

**Figure supplement 3.** Dinucleotides frequency of TBB positions.

**Figure supplement 4.** MNase-digestion correction for dinucleotides frequency of TBB positions.

control for the influence of MNase digestion bias, we simulated MNase digestion based on its observed sequence preference (*Gaffney et al., 2012*), and identified nucleosome positions from the in silico MNase digestion data set. We used these in silico nucleosome positions to normalize the bias that was merely a result of MNase digestion (*Figure 4—figure supplement 4*). After these corrections, we observed dinucleotide AA/AT/TA/TT and CC/CG/GC/GG motifs displaying 10 bp periodicity (*Figure 4C*), consistent with the dinucleotide frequency around the chemically determined nucleosome positions in yeast (*Brogaard et al., 2012*). Randomly shifting the determined TBB

nucleosome positions by an average of 2 bp greatly reduced the observed dinucleotide periodicity and shifting the positions by an average of 5 bp completely eliminated the periodic pattern (*Figure 4D*), demonstrating the importance of precisely mapping nucleosome positions in vivo.

Paired-end nucleosome sequencing has been used to map chromatin structure in higher eukaryotes (*Gaffney et al., 2012*), but a base-pair resolution nucleosome map has been lacking for organisms other than yeast. We applied the TBB approach to determine the positions of human nucleosomes from a recent paired-end MNase-seq study (*Gaffney et al., 2012*). The frequency of AA/AT/TA/TT and CC/CG/GC/GG motifs displayed clear 10 bp periodicity relative to human TBB nucleosome positions, which was comparable to the motif periodicity observed in yeast (*Figure 4A and B*, *Figure 4—figure supplement 2*). In contrast, when DNA sequences were aligned at the consensus centers of human nucleosomes, the frequency of these dinucleotide motifs did not exhibit a clear periodicity with respect to the consensus centers (*Figure 4—figure supplement 1*). With the template-based Bayesian approach, we demonstrated that nucleosome positions determined from MNase-based methods indeed present periodic dinucleotide frequency globally, and that this feature may be a conserved signal for nucleosome positioning in vivo.

## Heterogeneous nucleosome positions at the TSS

Positioning of nucleosomes over yeast transcription start sites (TSSs) is influenced by promoter sequence signals and *trans* acting factors (*Jiang and Pugh, 2009b*; *Radman-Livaja and Rando, 2010*; *Segal and Widom, 2009b*). This observation, together with our finding that overlapping positions are commonly observed in yeast and human chromatin, motivated us to examine the relationship between TSSs and overlapping nucleosome positions (resulting from alternative positions). We identified all TBB nucleosome positions that were closely associated with TSSs (*Jiang and Pugh, 2009a*) and organized them into groups consisting of either unique or overlapping TBB nucleosome positions for each gene (*Figure 5A and B*). If a group contained several overlapping TBB nucleosome positions, the center of this group was defined as the numeric average of these positions. All groups were aligned at their centers and ranked by the spacing between TBB nucleosome positions within the groups (*Figure 5C*). The TBB nucleosome positions exhibited a bimodal shape for at least three quarters of the genes with nucleosome-covered TSSs (*Figure 5C*, 'positions'), suggesting that alternative positioning is a common feature of nucleosomes near TSSs. The number of sequenced read midpoints showed the same bi-furcating shape and was highly consistent between two experiments (*Figure 5C*, 'T1 midpoints' and 'T2 midpoints'). The occupancy of nucleosome midpoints at each gene correlated well between two experiments (T1 and T2), indicating alternative nucleosomes can be reliably observed (*Figure 5D*). The distribution of read midpoints was symmetric along the direction of gene transcription (*Figure 5—figure supplement 1*), and could not be explained by the sequence bias of MNase digestion (*Figure 5C*). When we examined the midpoints attributed to each nucleosome position within each group (*Figure 5—figure supplements 1–2*), we found that most groups showed less than 2-fold variation in the read occupancy between alternative TBB nucleosome positions, suggesting that the different positions are significantly represented in the cell population and therefore are likely of biological relevance. These alternative positions were not a consequence of differential digestion at nucleosome edges (*Weiner et al., 2010*), as both ends of paired-end sequencing reads showed two peaks that match the edges of alternatively positioned nucleosomes (*Figure 5E*). Differential digestion at nucleosome edges would produce two peaks of read ends on one side and just one peak on the other read end (*Weiner et al., 2010*). Of the genes with their TSSs covered by nucleosomes, approximately one quarter contain more than two TBB nucleosome positions at TSSs, suggesting that chromatin structure at the transcription initiation site is complex and heterogeneous.

## Alternative chromatin states and transcription initiation

Alternative positions of nucleosomes around the TSS may allow the transcription start site of a gene to be exchanged between accessible and inaccessible states. To determine if this is the case, we analyzed the location of transcription start sites with respect to the alternative TBB nucleosome positions. We defined the nucleosome with its center position closer to the TSS as the 'proximal' nucleosome, and the nucleosome with its center position further away from the TSS as the 'distal' nucleosome (*Figure 6A*). The unique, proximal and distal nucleosomes were digested by MNase to

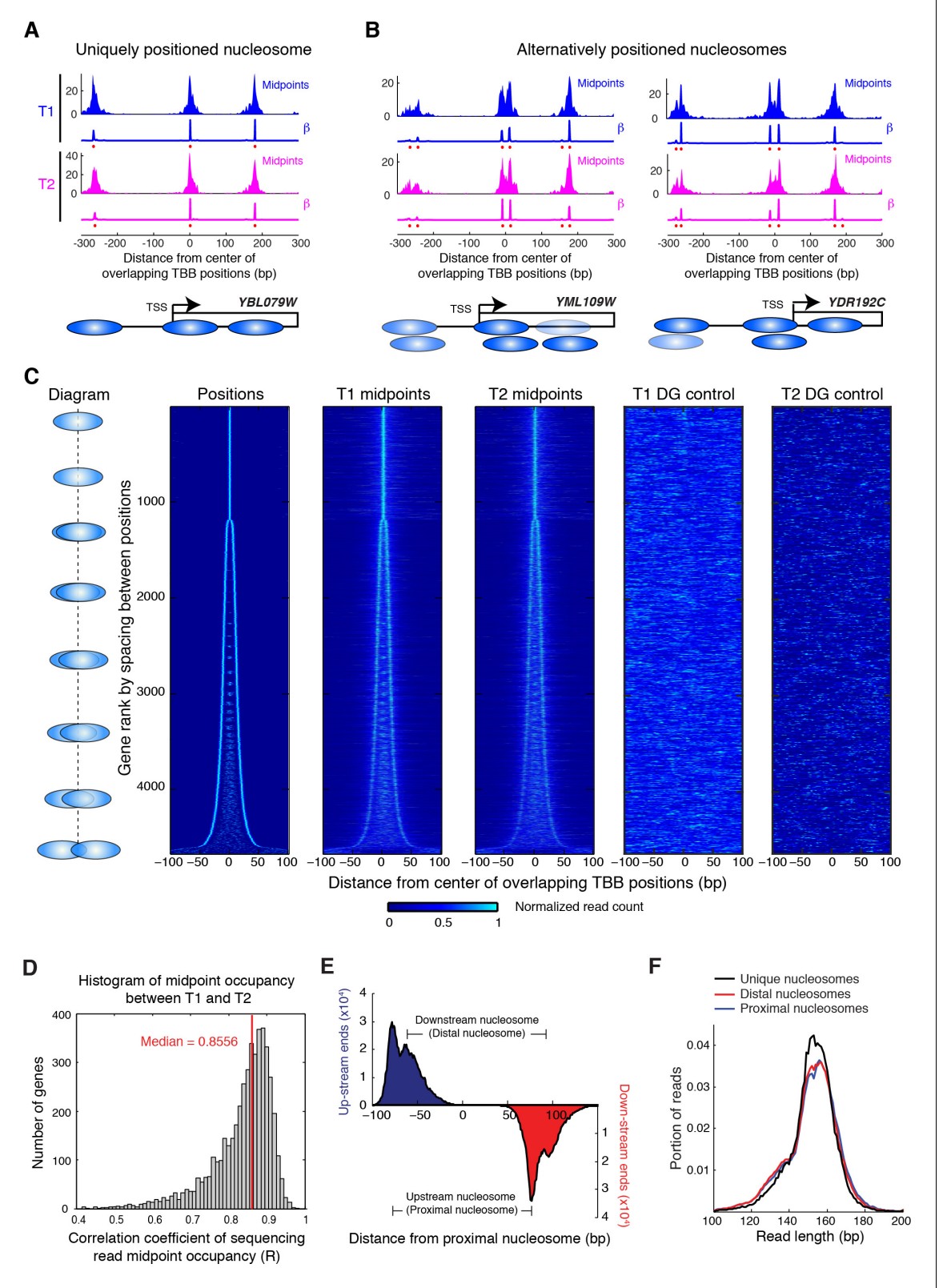

**Figure 5.** Alternatively positioned nucleosomes at transcription start sites. (**A,B**) Examples of a uniquely positioned nucleosome (**A**), and alternatively positioned nucleosomes at the TSS (**B**). Blue and magenta traces show the sequencing read midpoint occupancy and fitted coefficient β from experiments 'T1' and 'T2', respectively. Ovals indicate the TBB nucleosome positions and are colored based on coefficients β. (**C**) Heat map showing the TBB nucleosome positions ('Positions'), the occupancy of read midpoints from two experiments (T1 and T2), and the occupancy of read midpoints

*Figure 5 continued on next page*

*Figure 5 continued*

from the two MNase bias simulations ('T1 DG control' and 'T2 DG control'). All data are aligned by the centers of selected unique or alternative nucleosome positions (essentially the consensus center of +1 nucleosomes) that overlap with the transcription start site (TSS) (4672 open reading frame in total). The order of transcripts is ranked by the maximum space between TBB nucleosome positions within each group, as illustrated by the diagrams on the left. The positive direction of the x-axis indicates 5′ to 3′ for all transcripts. (D) Correlation coefficient of the sequencing read midpoint occupancy (un-smoothed) in experiment T1 and T2 for each gene in panel C. (E) Graph showing the end location of paired-end sequencing reads of the upstream and downstream nucleosomes in panel C (later defined as the proximal and distal nucleosomes, respectively). The ends are aligned at the position of upstream nucleosomes (proximal nucleosomes). (F) Length distribution of paired-end sequencing reads in the unique, proximal and distal nucleosomes at gene promoters. In both (E) and (F), if the midpoint of a sequencing read is within 5 bp of the position of a nucleosome, it is counted as a read of this nucleosome.

The following source data and figure supplements are available for figure 5:

**Source data 1.** Source data for *Figure 5C*, with arrays of positions, midpoints from T1 and T2 used for generating the graph.
**Figure supplement 1.** Reads occupancy between alternatively positioned nucleosomes.
**Figure supplement 2.** Heat map showing the TBB nucleosome positions and the midpoint read occupancy.

similar sizes, further suggesting that the alternative nucleosome positions are not a result of MNase digestion bias (*Figure 5F*).

Among uniquely positioned nucleosomes, TSSs weree enriched ~10 bp inside the boundary of the nucleosome core (*Figure 6A*, magenta curve). Among the alternatively positioned nucleosomes, the proximal nucleosomes were located much closer to the TSS where the nucleosome core restricts the accessibility of these factors and therefore competes for active transcription (*Figure 6A*, cyan curve). For the distal nucleosomes, TSSs were aligned near the edge of the nucleosome core (*Figure 6A*, red curve), making these sites and the upstream regulatory elements accessible to other DNA binding factors. Alternatively positioned nucleosomes at TSSs gave rise to different accessibilities of the promoter elements within a population of cells, which may reflect the different activities in gene transcription. If one focused only on the average centers of these nucleosomes (essentially the consensus centers of the +1 nucleosomes), one would draw the conclusion that nucleosomes occlude the TSSs of most genes (*Figure 6A*, black line).

To explore how alternative TBB nucleosome positions might influence transcription, we focused on the initial steps of this process: the assembly of the transcription pre-initiation complex (PIC), which generally requires binding of TBP (TATA-binding protein); and the subsequent recruitment of general transcription factors and RNA polymerase II (Pol II) (*Buratowski et al., 1989*; *Green, 2000*; *Orphanides et al., 1996*; *Roeder, 1996*). Recently, it was suggested that the +1 nucleosome downstream of the TSS assists in the assembly of the PIC (*Rhee and Pugh, 2012*). However, the binding of the PIC extends into the boundary of the +1 nucleosomes (*Rhee and Pugh, 2012*), presenting a physical conflict between PIC binding and nucleosomes. To test if the resolution of alternatively positioned nucleosomes would provide a better understanding of the interplay between nucleosomes and the assembly of the PIC, we analyzed the occupancy of the PIC (determined with the ChIP-exo method (chromatin immunoprecipitation with lambda exonuclease digestion) by Rhee and Pugh [*Rhee and Pugh, 2012*]) around the identified distal and proximal nucleosomes, respectively. Most subunits of the PIC, including TBP, TFIIA, TFIIB, TFIIE, TFIIF, TFIIH and TFIIK, showed peak enrichment at the edge of the distally positioned nucleosome (*Figure 6B*, cyan), suggesting that the +1 nucleosome in the distal position marks a boundary for the assembly of the PIC. This boundary may be provided by the positioned nucleosomes or by binding of one or more components of the transcription complex. In contrast, the proximally positioned nucleosome covered most of the regions enriched for PIC binding (*Figure 6B*, red), suggesting that the +1 nucleosome in the proximal position is not compatible with the binding of transcription machinery. Intriguingly, TATA-box and TATA-like sequences were enriched in front of the proximal nucleosomes (*Figure 6C*), suggesting that proximal nucleosomes do not block recognition of most TATA elements. Thus, the proximal nucleosome may represent a promoter state that is not associated with active transcription and is populated prior to the assembly of the PIC (*Figure 6D*, Proximal state). The distal nucleosome may

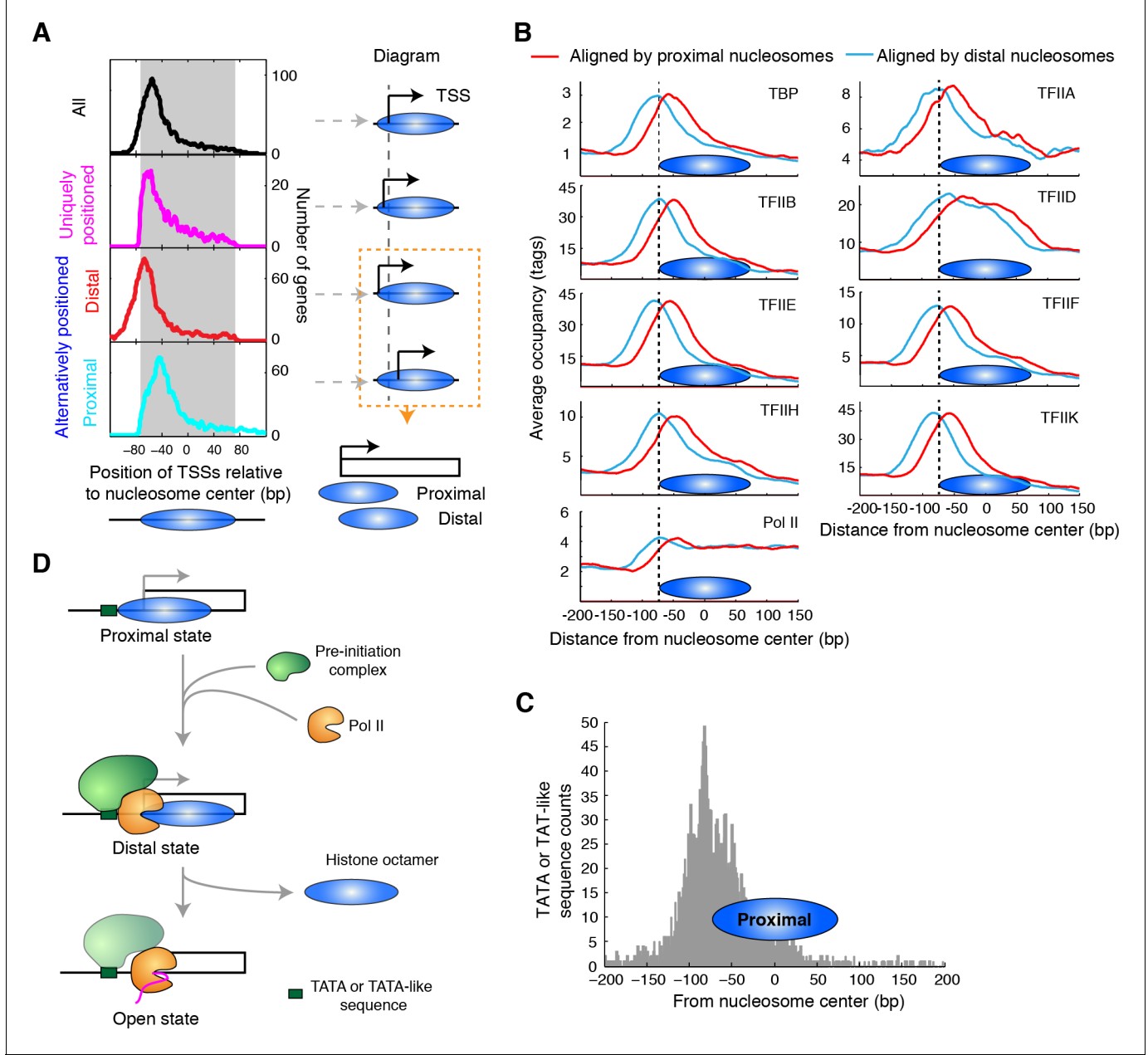

**Figure 6.** Alternatively positioned nucleosomes and transcription pre-initiation complex. (**A**) Plots (left) showing the locations of TSSs relative to the average centers of all overlapping TBB nucleosome positions shown in *Figure 3C* ('All', black) to uniquely positioned nucleosomes (magenta) and to the distally (red) and proximally (cyan) positioned nucleosomes. The gray area marks the region covered by the nucleosome core. The cartoon diagrams on the right illustrate the location of the TSS relative to the nucleosome dyad. (**B**) Area showing the average occupancy of subunits of the pre-initiation complex (TBP, TFIIA, TFIIB, TFIID, TFIIE, TFIIF, TFIIH, TFIIK and RNA polymerase II (Pol II); determined by ChIP-exo) (*Rhee and Pugh, 2012*) aligned at the center of the proximal (cyan) and the distal nucleosomes (red), determined by the TBB approach. (**C**) Bar graph showing the distribution of TATA box and TATA-like sequences of all genes (*Rhee and Pugh, 2012*) aligned at the dyad of the proximal nucleosomes. (**D**) Illustration of the 3-step model for transcription initiation mediated by alternatively positioned nucleosomes.

The following source data is available for figure 6:

**Source data 1.** Source data for *Figure 6*, with genes and coordinates of TSSs, unique, proximal and distal nucleosome positions.

represent a promoter state in the process of transcription initiation (*Figure 6D*, Distal state). Once transcription begins, the barrier of nucleosomes no longer hinders Pol II as its occupancy extends through the region covered by the +1 nucleosome (*Figure 6B*).

The location of uniquely positioned nucleosomes relative to TSSs was similar to that of the distal nucleosomes (*Figure 6A*, magenta and red curves), suggesting that the binding and assembly of PIC may encounter less competition from chromatin at these genes. We found that genes with uniquely positioned nucleosomes at their TSSs were enriched in highly expressed genes (*Newman et al., 2006*) (top 500 genes, p = 0.001, Fisher's exact test). More strikingly, we found that 73% of the genes encoding ribosomal proteins had TSSs with uniquely positioned nucleosomes (p<0.00001, Fisher's exact test), highlighting a connection between the configuration of nucleosomes (unique vs alternative) and gene expression.

## Sequence determinants of unique and alternative chromatin states

DNA sequence features, such as periodic occurrence of dinucleotides (AA/AT/TA/TT or CC/CG/GC/GG) and consecutive dA or dT sequences (poly(dA:dT)), can influence nucleosome positions (*Struhl and Segal, 2013*). We therefore tested the possibility that DNA sequence determinants may underlie the difference between uniquely, proximally, and distally positioned nucleosomes around TSSs. The frequency of dinucleotides occurred periodically ~10 bp from the positions of either proximal or distal nucleosomes and resembled the periodic feature of nucleosomes in yeast genome ($R^2$ = 0.75, *Figure 7A*). This periodicity was less pronounced in the uniquely positioned nucleosomes ($R^2$ = 0.40 between dinucleotide frequency in uniquely positioned nucleosomes and all nucleosomes in yeast genome; *Figure 7A*). Autocorrelation analysis of the dinucleotide frequency revealed a lack of periodicity in DNA sequences around the uniquely positioned nucleosomes, further highlighting the difference between uniquely and alternatively positioned nucleosomes (*Figure 7B*).

Poly(dA:dT) tracks are important for nucleosome depletion and correlate with transcriptional activity (*Hughes et al., 2012*; *Iyer and Struhl, 1995*; *Raveh-Sadka et al., 2012*; *Segal and Widom, 2009a*; *Struhl, 1985*). When we examined the occurrences of poly(dA:dT) sequences at least 6 nucleotides in length, we found that the uniquely positioned nucleosomes were flanked by a strong enrichment of poly(dA:dT)$_6$ sequences (*Figure 7C*). A similar but weaker enrichment of poly(dA:dT)$_6$ sequences was also observed flanking both the proximally and the distally positioned nucleosomes (*Figure 7D*). In comparison, no enrichment of poly(dA:dT)$_6$ sequences was observed in randomly selected DNA sequences (*Figure 7C and D*, gray curve). Overall, uniquely positioned nucleosomes and alternatively positioned nucleosomes presented different DNA sequence features, suggesting that these sequence determinants may contribute to the different configurations of chromatin states at gene promoters.

## Discussion

Knowing the precise location of nucleosomes in vivo is key to understanding how diverse biological processes are regulated (*Jiang and Pugh, 2009b*). Although MNase-seq methods have been widely used for mapping nucleosomes in many organisms, the commonly reported consensus centers of nucleosomes do not precisely reflect the location of nucleosomes with respect to DNA sequence. We developed a new computational method to accurately determine the positions of nucleosomes at base pair resolution based on paired-end sequencing data, providing a general and accessible approach to accurately map the basic chromatin structure in eukaryotes.

The periodicity in the frequency of dinucleotide AA/AT/TA/TT and CC/CG/GC/GG motifs has been observed in vivo in several organisms, including unicellular fungi (*Albert et al., 2007a*; *Brogaard et al., 2012*), insects (*Mavrich et al., 2008*), chicken (*Satchwell et al., 1986*) and human (*Gaffney et al., 2012*), and was thought to strongly influence the positioning of nucleosomes (*Kaplan et al., 2009*; *Struhl and Segal, 2013*). However, this feature was only obvious when the sequences of nucleosomal DNA fragments exactly 147 bp in length were selected for analysis (*Albert et al., 2007a*; *Gaffney et al., 2012*; *Mavrich et al., 2008*; *Segal et al., 2006*), and was much weaker or even disappeared around the consensus centers of nucleosomes that were determined with an entire data set (*Brogaard et al., 2012*; *Valouev et al., 2011*). Thus, differences in the selection and analysis of MNase-seq data may result in different conclusions about nucleosome positioning in vivo (*Gaffney et al., 2012*; *Valouev et al., 2011*). With the high-precision nucleosome

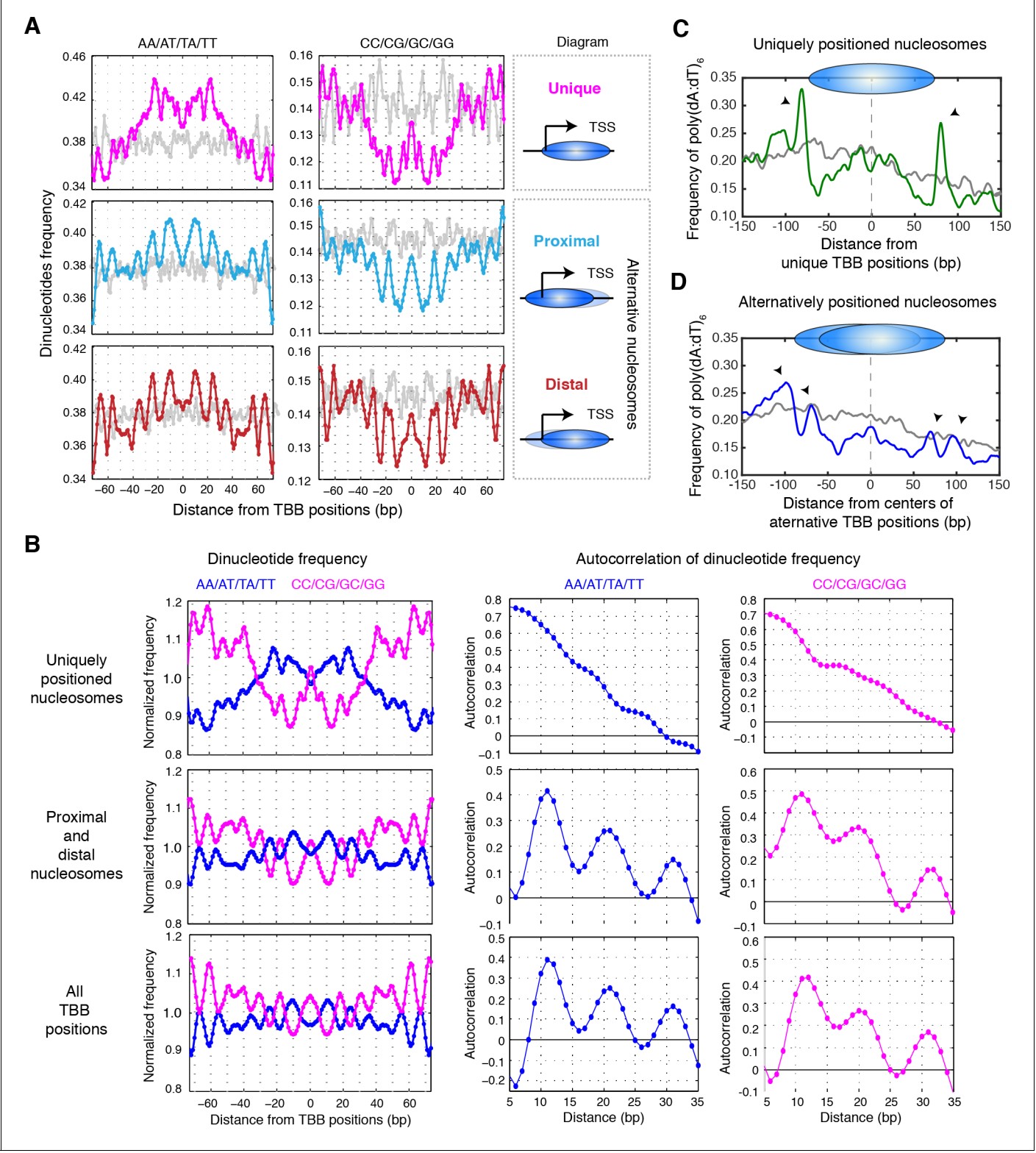

**Figure 7.** Sequence features of uniquely and alternatively positioned nucleosomes. (**A**) Plots showing the frequency of AA/AT/TA/TT and CC/CG/GC/GG dinucleotides of DNA sequences aligned at the TBB nucleosome positions of either unique, proximal or distal nucleosomes at gene promoters (illustrated by the diagrams). (**B**) Plots showing the normalized dinucleotide frequency (smoothed with a 3 bp window) of DNA sequences aligned at unique nucleosomes, alternative nucleosomes (proximal + distal) at gene promoters or all TBB nucleosome positions, and the autocorrelation analysis (performed in MATLAB) of the dinucleotide frequency within nucleosome core (−73–73 bp). (**C,D**) Plots showing the frequency of poly(dA:dT)$_6$

*Figure 7 continued on next page*

*Figure 7 continued*

sequences aligned at unique nucleosomes or the average centers of alternatively positioned nucleosomes. The positive direction of the x-axis indicates 5′ to 3′ for all transcripts. Black arrows mark the enriched poly(dA:dT)$_6$ signals. Overall, 1172 genes contain unique nucleosomes and 3469 genes contain proximal and distal nucleosomes at their promoters. Gray traces present the analysis for a random permutation control of selected promoters and the location of nucleosome positions (**A**,**C**,**D**). The number of these random locations matches the number of nucleosomes in each plot.

position maps generated by TBB, we directly demonstrated that the DNA sequences around the mapped nucleosome positions exhibited 10 bp periodicity in the frequency of dinucleotide motifs in both yeast and human cells, arguing that this sequence feature is a universal signal for positioning nucleosomes in vivo at the whole genome level.

Positioning of nucleosomes is dynamic and linked to many biological processes. Identifying overlapping nucleosome positions that cannot co-exist on the same DNA molecule at the same time provides a snapshot of possible chromatin states in dynamic processes. We observed at least two states of chromatin where gene transcription is initiated: a state where a nucleosome covers the transcription start site and competes with transcription machinery, but leaves the core promoter sequences mostly accessible ('proximal state'); and another state where a nucleosome barely covers the transcription start site and seems poised for the assembly of the pre-initiation complex ('distal state'). We propose a 3-step model (*Figure 6D*) to describe the transcription initiation steps influenced by nucleosomes, including a state where RNA polymerase transcribes through the nucleosome covered sequence ('open state'). The transition from the 'proximal state' to the 'distal state' and to the 'open state' could be actively driven by the assembly of the PIC; in this case, different nucleosome states may reflect rate-limiting steps during transcription initiation. Alternatively, this transition could be controlled by transcription factors, chromatin modulators and even the preference of DNA sequences as a means to regulate and assist gene transcription. Interestingly, we found that TSSs of highly expressed genes are enriched with uniquely positioned nucleosomes, which resemble nucleosomes in their 'distal state'. It is possible that the +1 nucleosomes of these most highly expressed genes are oriented in a way that allows constitutive assembly of the PIC and therefore frequent transcription activity. Further genomic analysis and mechanistic studies will shed light on the generalization and the underlying principles of this model.

DNA sequences may play a role in determining whether nucleosomes have unique or alternative positions. Unique nucleosome positions may result from the combination of the strong nucleosome boundary signal from poly(dA:dT) sequences and the lack of a rotational positioning signal from periodic dinucleotide motifs. Conversely, alternatively positioned nucleosomes may be a consequence of a weak boundary signal and a strong rotational positioning signal that allows nucleosomes to interconvert between multiple equivalent positions. The exact positions of these nucleosomes and how alternative nucleosomes are positioned relative to each other may be further controlled by other factors associated with promoters, such as chromatin modifiers, remodelers and transcription factors. The configurations of nucleosomes (unique vs. alternative) may influence the transcription of a gene; if promoter DNA sequences contain information that helps to specify the nucleosome configurations, eukaryotic genome could use nucleosome configurations as a means to encode transcriptional programs.

It is worth noting that the template-based Bayesian approach does not capture all possible positions of nucleosomes within the cell population – it identifies the nucleosome positions that are represented with strong statistical confidence (Materials and methods), reflecting the different configurations of nucleosomes that are likely to be relevant in biological processes. Revealing these heterogeneous configurations of nucleosomes in cells will provide new insights into how chromatin regulates or is influenced by biological processes. Furthermore, mutually exclusive positions of protein-DNA complexes can imply dynamic information. As an example, the alternatively positioned nucleosomes at TSSs suggest a change in chromatin structure during transcription initiation. Similar analytic approaches can be applied to other systems to reconstruct dynamic processes.

## Materials and methods

### Nucleosomal DNA preparation

Detailed conditions of cell culture, chromatin isolation, MNase digestion and DNA extraction have been previously described (*Zhou and O'Shea, 2011*). Briefly, 100 OD$_{600}$ units of yeast cells (EY57; genotype K699, *ade2-1 trp1-1 can1-100 leu2-3,112 his3-11,15 ura3 GAL+*) growing in synthetic complete medium at 30°C with shaking were harvested at OD$_{600}$ ~0.5 for each experiment. Cells were crosslinked with 1% formaldehyde for 15 min at room temperature and quenched with 125 mM glycine for 5 min. Collected cells were lysed mechanically with glass beads at 4°C with Mini-Bead-beater-24 (Biospec) in lysis buffer (50 mM HEPES, pH 7.5, 150 mM NaCl, 2 mM EDTA, 1% Triton X-100, 0.1% Sodium-Deoxycholate) and the chromatin pellet fraction was digested with either 0.5, 1, 2 or 4 U MNase (Worthington Biochemical) at 37°C for 30 min. DNA fragments corresponding to mono-nucleosomes were extracted from an agarose gel after electrophoretic separation.

### Sequencing library preparation and read mapping

For each experiment, 0.5 μg of mono-nucleosomal DNA (~5 pmol) was used for sequencing library construction according to the Illumina Truseq paired-end protocol. ~1 pmol of paired-end ligated library was recovered from the agarose gel for each experiment before PCR amplification. Since the amount of ligated library was sufficient for sequencing, we only performed 4 cycles of PCR amplification with the Illumina primer cocktail to obtain a double stranded sequencing library. The number of sequencing clusters generated on the flow-cell represented at most 1/10,000 of the amplified library, such that the chance was very small to have more than one copy of duplicated library molecules clustered on the flow-cell (p<1.8 × 10$^{-5}$, binomial distribution). Therefore, we treated each paired-end read as an individual nucleosome from a single cell. Libraries were sequenced with an Illumina Hiseq 2000 and the paired-end sequencing reads were aligned to the *S. cerevisiae* genome Scer02 with bowtie 2.0 (*Langmead et al., 2009*). We mapped reads with two or fewer mismatches, and insert lengths were restricted to sizes between 100 bp and 300 bp. Reads with multiple reportable alignments were randomly assigned to one such alignment. The midpoints of all sequenced DNA fragments were randomly rounded to an integer genomic coordinate and used to generate sequencing read midpoint occupancy maps.

We obtained 33.7 million and 34.3 million mappable paired-end reads for the T1 and T2 experiments, respectively. All analyses presented in this work were done with the T1 dataset unless specified.

### Detection of consensus centers and TBB nucleosome positions

The consensus centers of nucleosomes were determined based on the fitted coefficient β with a standard Parzen window peak calling method (*Tsankov et al., 2010*). TBB nucleosome positions were determined at FDR ≤ 0.05, where the null distributions of nucleosome positions were estimated from randomly sampled nucleosome reads distributed based on the local read density. We developed two different null distributions: a random null distribution where simulated sequencing reads were assumed to be uniformly distributed within each region, and an MNase digestion-aware null where the simulated sequencing reads were sampled to match the observed dinucleotide end frequency of the paired-end reads. We applied the digestion-aware null to identify TBB nucleosome positions with a threshold of FDR ≤ 0.05, resulting in 62,035 consensus centers and 120,347 TBB nucleosome positions in the T1 experiment, and 61,532 consensus centers and 129,505 TBB nucleosome positions in the T2 experiment. When we attempted to identify possible nucleosome positions that were entirely due to the sequence bias of MNase digestion, we analyzed the simulated MNase digestion data though our TBB pipeline and applied the random null distribution with a threshold of FDR ≤ 0.05. This allowed us to identify 49,739 and 50,076 positions to analyze and correct for influence of MNase digestion bias.

### Human nucleosome analysis

Analyses on human nucleosomes were performed using paired-end sequencing reads with length between 126 bp and 184 bp (*Gaffney et al., 2012*). We randomly selected a 10 Mbp region on human chromosome 12, position 38,000,000 bp to 48,000,000 bp, to test the TBB approach. In total,

we identified 43,991 consensus centers of nucleosomes and 251,948 TBB nucleosome positions in this selected region. All the identified positions were used in our analysis.

## Dinucleotide frequency analysis

All TBB nucleosome positions determined in T1 were used for the dinucleotide frequency analysis unless specified. DNA sequences from 100 bp upstream to 100 bp downstream of each TBB nucleosome position were aligned and the frequency of AA/AT/TA/TT and CC/CG/GC/GG dinucleotides was calculated for every base pair. This calculated di-nucleotide frequency was then normalized to the average dinucleotide frequency and presented in the plots.

For the permutation of TBB nucleosome positions, we first randomly selected 62,035 TBB nucleosome positions (the same number as consensus centers of nucleosomes). For each of these positions, we randomly shifted the genome coordinate of the position by 0–4 bp with an overall average of 2 bp, or by 0–10 bp with an overall average of 5 bp. The frequency of AA/AT/TA/TT dinucleotides with respect to these permutated positions is presented in the plot *Figure 4D*, in comparison to that of the original TBB nucleosome positions.

## Identification of overlapping nucleosomes around TSSs

To determine if the observed overlapping TBB nucleosome positions are biologically meaningful, we excluded TBB nucleosome positions whose occupancy is less than 4.8 reads (about half of the median sequencing read occupancy of all TBB nucleosome positions), which account for ~5% of the detected TBB nucleosome positions. The position occupancy was calculated as the average read count within a 3 bp window centered at TBB nucleosome positions. We then combined individual TBB nucleosome positions that are separated by less than 80 bp into groups of overlapping nucleosomes (overlap by ~ half of a nucleosome). In total, 57,643 groups containing either unique or overlapping TBB nucleosome positions were identified in the yeast genome. Groups that contain more than 9 nucleosome positions (139 groups, 0.24%) were further excluded from this analysis because they likely reside in regions with delocalized nucleosomes. The remaining groups were compared to the positions of TSSs of all ORFs in yeast genome (*Jiang and Pugh, 2009a*), and 4641 were identified to overlap with TSSs (TSSs within 73 bp distance of any TBB nucleosome position in the group). All 4641 were analyzed and presented in *Figure 5C*. 3307 of these groups contain no more than 2 TBB nucleosome positions and their analyses are shown in *Figure 5—figure supplement 2*.

## PIC binding and TATA-box/TATA-like sequences

The occupancy data for the subunits of the PIC was generated from the supplementary data of the work of Rhee and Pugh (*Rhee and Pugh, 2012*). The genome coordinates for PIC occupancy were calculated based on the positions of the +1 nucleosomes kindly provided by Ho Sung Rhee.

Both TATA-box sequences and TATA-like sequences identified by Rhee and Pugh (*Rhee and Pugh, 2012*) were analyzed in this study and presented in *Figure 6C*. However, a substantial fraction of the TATA-like sequences are located next to or even after the annotated transcription start site – TATA-like sequences of ~600 genes are within 5 bp of TSSs. If we exclude these genes from our analysis, the majority of the TATA-box/TATA-like sequences are located outside the boundary of the proximal nu cleosomes.

## Nucleosome digestion variability template estimation

We estimated the digestion variability template from the distribution of sequencing read lengths. This digestion variability reflects a combination of factors that influences the size of protected nucleosomal DNA, such as over- or under- digestion of intact nucleosomes. This estimation was based on a simple, symmetric model of MNase digestion for each sequenced fragment. Using this model, we obtained a nonparametric maximum likelihood estimator of the distribution of digestion variability ($e$) at each end of a sequenced fragment ($e_{1j}$, $e_{2j}$). Therefore, we can infer the digestion variability template as the distribution of the variability in paired-end read midpoints due to digestion. We denoted the length of each paired-end read $j$ as $\ell_j$ and assumed

$$\ell_j = \ell_0 + e_{1j} + e_{2j}, \qquad e_{1j}, e_{2j} \sim IID.$$

$\ell_0$ is defined as the length of DNA covering the histone core octamer (147 bp) (*Kornberg and Lorch,*

*1999*) and the $e_{1j}$ and $e_{2j}$ terms are the digestion errors at each end of the read $j$. We assumed MNase has the same propensity towards over-or under-digestion at each end on average; therefore, we modeled these errors as bounded and symmetric between the ends of sequencing reads. Along the genome, the distributions of digestion errors at the ends of each read are mirror images of each other, but the two errors of the same read were independent. So positive values of $e_{1j}$ or $e_{2j}$ imply that DNA sequence not covered by the histone core octamer was left at the ends after MNase digestion, and negative values imply that MNase over-digested the nucleosomes.

Under this model, each sequencing read's midpoint varies around its nucleosome's center according to the distribution of $d_j \equiv \frac{1}{2}\left(e_{1j} - e_{2j}\right)$. The template $\vec{t}$ specifies this distribution, expressed in vector form and transformed to account for the random rounding of fragment centers to integer positions. Hence,

$$t_k = P\left(d_j = k\right) + \frac{1}{2}\left(P\left(d_j = k - \frac{1}{2}\right) + P\left(d_j = k + \frac{1}{2}\right)\right)$$

For $k = -w, \ldots, w$, yielding a vector $\vec{t}$ of length $2w + 1$. To setup the estimation task, we defined two probability distributions,

$$\begin{aligned} p(i) &= \Pr(l_j = i), \\ q(i) &= \Pr(e_{1,j} = 1). \end{aligned}$$

It is known that $l_j \geq 0$, which implies $e_{1,j}, e_{2,j} \geq -\lfloor \frac{l_0}{2} \rfloor$. If the longest observed fragment length is $l_{max}$, we also have $Pr(l_j > l_{max}) = 0$. The non-parametric Maximum Likelihood Estimation (MLE) also enforces $l_j \leq l_{max}$, which implies $e_{1,j}, e_{2,j} \leq l_{max} - l_0 + \lfloor \frac{l_0}{2} \rfloor$. We could then write

$$p(i) = \sum_{k=-\lfloor \frac{l_0}{2} \rfloor}^{l_{max}-l_0+\lfloor \frac{l_0}{2} \rfloor} q(k)q(i - l_0 - k).$$

The resulting log-likelihood for the observed fragment lengths is

$$\ell(q) = \sum_{j=1}^{M} \log p\left(l_j\right).$$

We maximized $l(q)$ numerically, using a multivariate logit transformation on the values $q(k)$ to avoid bounded optimization. Using the L-BFGS algorithm (*Zhu et al., 1997*) on a laptop with a Core i5 processor and 8GB of RAM, this maximization requires ~40 s for a typical experiment. The complexity of this computation depends only on the number of distinct read lengths observed (i.e., 151 bp, 152 bp, etc).

We obtained the template distribution $\vec{t}$ from $q$ via a convolution sum and linear transformation. We first obtained the distribution of $e_1 - e_2$ via

$$u(i) = P(e_1 - e_2 = i) = \sum_{k=-\lfloor \frac{l_0}{2} \rfloor}^{l_{max}-l_0+\lfloor \frac{l_0}{2} \rfloor} q(k)q(k - i).$$

We finally transformed the distribution $u(i)$ to the desired template $t(i)$ by accounting for random rounding, as

$$t(k) = \frac{1}{2}u(2k - 1) + u(2k) + \frac{1}{2}u(2k + 1).$$

The resulting digestion variability template accurately reflects both variation from enzymatic digestion and the details of the preprocessing.

## Segmentation of chromosome for nucleosome position estimation

Each chromosome was segmented into disjoint, contiguous regions with similar statistical properties (sequencing coverage). This was done to control for local variation in sequencing coverage and

nucleosome occupancy, allowing for efficient estimation of local structure with minimal sensitivity to distant observations. To accomplish this, we started from ORFs and refined these natural demarcations into a statistically useful segmentation. We first enumerated all open reading frames (ORFs) and intergenic regions on each given chromosome. Merging overlapping ORFs into single segments yielded a starting set of contiguous, non-overlapping segments. We then iteratively merged the most similar short segments until all segments exceed a minimal length (800 bp). The similarity was defined based on sequencing read midpoints per base pair within each segment. This yielded a segmentation for which each segment has sufficient length to estimate local properties of the nucleosome position distribution. For the *S. cerevisiae* genome, we obtained a total of 5135 segments for our T1 experiment with a mean segment length of 2351 bp (minimum length of 800 bp and maximum length of 14,499 bp). The results on the T2 experiment were extremely similar with 5118 segments and a mean length of 2359 bp. This segmentation was fixed and used in all subsequent computation. The region of human chromosome 12, position 38,000,000 bp to 48,000,000 bp was segmented based on the annotated introns and exons of human genome hg18 (Roche, NimbleGen). Other segmentation techniques based on the biological structure of each chromosome can be used and may be more appropriate for mammalian genomes.

## Computational model and notation

Our probabilistic model for the observed read midpoints captures the variation due to nuclease digestion as well as the variation from biological sources (for example, biological heterogeneity among cells, stochastic fluctuation in the selection of fragments for sequencing, and other factors in sample preparation and analysis). To accomplish this, we extended discrete convolution models from signal processing to account for the particular types of structure and variation found in MNase-seq data. For each chromosome, we defined $y_k$ as the number of sequencing read midpoints aligned to position $k$, where $k$ ranges from 1 to $N$ (the length of the chromosome). Given the segment $s : \{1...N\} \longrightarrow \{1...S\}$, which maps the $N$ base pair locations to $S$ segments on this chromosome, we posited

$$y_k \mid \lambda_k \sim Poisson(\lambda_k)$$

$$\vec{\lambda}_{(N \times 1)} \equiv X_{(N \times (N - 2\lfloor \ell_0/2 \rfloor))} \vec{\beta}_{((N - 2\lfloor \ell_0/2 \rfloor) \times 1)}$$

$$\beta_k > 0 \; for \; k = \lfloor \ell_0/2 \rfloor + 1 \ldots N - \lfloor \ell_0/2 \rfloor$$

$$log\beta_k \sim Normal\left(\mu_{s_k}, \sigma^2_{s_k}\right)$$

where $X$ specifies the contribution of a nucleosome positioned at $k$ to the expected number of reads at position $\mathrm{m}$ due to digestion variability, and $s_k$ is the region corresponding to position $k$. $X$ was defined as the matrix generated by the convolution of $\vec{t}$ with a sequence of length $N$. $\ell_0$ stands for 147 bp as mentioned before.

To complete the model specifications, we placed priors on $\mu_s$ and $\sigma^2_s$. We use independent conjugate priors for $\sigma^2_s$, assuming $\frac{1}{\sigma^2_s} \sim \mathrm{Gamma}(\alpha_0, \gamma_0)$. Our priors for $\mu_s$ are fully conjugate and independent across segments; we assumed $p\left(\mu_s \mid \sigma^2_s\right) \sim N\left(\mu_0, \frac{\sigma^2_s}{n_s \tau_0}\right)$ where $\mathrm{n}_s$ is the length of segment $s$.

## Estimation of nucleosome position signal

To estimate $\beta$ and detect individual nucleosome positions, we sampled from the posterior distribution of $\vec{\beta}$ using a Markov chain Monte Carlo (MCMC) sampler. By iterating through a carefully engineered sequence of random draws, this sampler obtains approximate samples from the given posterior. These draws of $\vec{\beta}$ form the basis for all of our subsequent summaries, particularly the detection of nucleosomes positions (using selected posterior probabilities) and the estimation of consensus positions (using selected posterior expectations). This sampler consisted of two alternating updates. At each iteration $r$, our algorithm

1. Drew $\left(\vec{\mu}^{(r)}, \vec{\sigma}^{2(r)}\right) \mid \vec{\beta}^{(r-1)}$ directly, then

2. Updated $\vec{\beta}^{(r)} \mid \left(\vec{\mu}^{(r)}, \vec{\sigma}^{2(r)}\right)$ via a distributed Hybrid Monte Carlo (HMC) step.

The first is a standard conjugate normal update, given the log-normal model for $\vec{\beta}$, and operates independently across segments. Full details can be found in *Blocker and Airoldi (2016)*.

The conditional posterior of $\vec{\beta}^{(t)} \mid \left(\vec{\mu}^{(t)}, \vec{\sigma}^{2(t)}\right)$ is not part of any standard family, so we turned to Hamiltonian Monte Carlo (HMC). The dimensionality of $\vec{\beta}$ makes a single HMC update for the entire vector both computationally infeasible and numerically unstable. For efficient computation, we leveraged the conditional independence structure of this conditional posterior. Subvectors of $\vec{\beta}$ separated by at least $2w$ entries are conditionally independent given $\left(\vec{\mu}^{(t)}, \vec{\sigma}^{2(t)}\right)$ and the entries of $\vec{\beta}$ between them. Consider the subvectors $\vec{\beta}_{[j_1:j_2]}$ and $\vec{\beta}_{[k_1:k_2]}$, with $j_1 < j_2 < k_1 < k_2$. The elements of $\vec{\beta}_{[j_1:j_2]}$ only affect $\vec{\lambda}_{[j_1-w:j_2+w]}$, and the elements of $\vec{\beta}_{[k_1:k_2]}$ only affect $\vec{\lambda}_{[k_1-w:k_2+w]}$. Hence, if $k_1 2 + 2w$, then $\vec{\beta}_{[j_1:j_2]}$ and $\vec{\beta}_{[k_1:k_2]}$ are conditionally independent given $\vec{\mu}$ and $\vec{\sigma}^2$.

We first fixed the length of each subvector that will be updated via a single HMC step to $B > 4w$. Next, considered two partitions of $\vec{\beta}$ into subvectors:

$$\vec{\beta}_{[1:B]}, \vec{\beta}_{[B+2w+1:2B+2w]}, \ldots, \vec{\beta}_{[n_b(B+2w)+1:N]},$$
$$\vec{\beta}_{[B/2+1:3B/2]}, \vec{\beta}_{[3B/2+2w+1:5B/2+2w]}, \ldots, \vec{\beta}_{[n_b(B+2w)B/2+1:N]}.$$

Within each partition, the subvectors are conditionally independent, and the union of these partitions includes all entries of $\vec{\beta}$.

For each iteration of our sampler, we cycled through each of these partitions, updating each subvector of $\vec{\beta}$ with an HMC step. As each subvector within each partition is conditionally independent, we can execute all HMC steps in parallel for each partition. This allows us to distribute the computational burden over hundreds of CPUs. Each of these distributed HMC steps is computationally efficient, as the log-conditional posterior's value and gradient can be computed via a convolution, lowering the computational cost to $o(B \log B)$ with the fast Fourier transform. A Python implementation of the sampler is available on GitHub, http://www.github.com/airoldilab/cplate.

## Estimation of consensus centers of nucleosomes

We defined consensus centers of nucleosomes as the average centers of possible overlapping nucleosomes within a small region. Each such position represents a cluster of nearby nucleosome positions and is similar to a 'nucleosome' obtained by standard methods (*Albert et al., 2007*; *Shivaswamy et al., 2008*; *Tirosh, 2012*; *Tsankov et al., 2010*). Based on the nucleosome position distribution $\vec{\beta}$, we determined the consensus centers of nucleosomes as the centers obtained from a standard Parzen window peak calling method (*Tirosh, 2012*). We smoothed each draw of $\vec{\beta}$ obtained from our sampler with a Gaussian window with a standard deviation of 20 bp. We then averaged these smoothed maps across draws and performed a greedy search for local maxima, enforcing a minimal spacing constraint of 147 bp. The comparisons presented in the main article are based on the standard practice of applying the same smoothing and search procedure directly to the midpoint counts (*Albert et al., 2007*; *Shivaswamy et al., 2008*; *Tsankov et al., 2010*).

## Detection of nucleosome positions

We defined nucleosome positions as a position along the genome with a greater chance observing a nucleosome than expected under a locally uniform distribution of sequencing read midpoints. With this definition in hand, we used carefully selected posterior probabilities (estimated using the above

algorithm) to find those positions with strong support from the observed data. Formally, to detect individual nucleosome positions, we first defined

$$C_{p,l}(k) = \frac{\sum_{i=-p}^{p} \beta_{k+i}}{\sum_{i=-l}^{l} \beta_{l+i}}$$

for each position $k$. We then estimated $P\left(C_{p,l}(k) > (2p+1)/(2l+1) \mid \vec{y}\right)$ for each $k$ using the MCMC sampler described previously. We then calibrated these Bayesian summaries against those implied by a null distribution of nucleosome positions within each region, constraining the number of reads within each segment to match the number observed and using information on the digestion variation in the form of the estimated template. We approximated this null distribution by repeatedly randomly permuting the observed reads within each segment.

In order to generate the Bayesian summaries implied by the null distribution described above, we then ran the proposed MCMC sampler on paired end sequencing reads drawn from the null, using the template and segmentation estimated from the observed data. From the sampler's output, we obtained an estimate of the distribution of $P\left(C_{p,l}(k) > (2p+1)/(2l+1) \mid \vec{y}\right)$ over positions $k$ under the null. We compared this to the distribution of posterior probabilities for the observed data and set a detection threshold to control the FDR using the method of Storey and Tibshirani (*Storey and Tibshirani, 2003*). For the datasets analyzed in this work, a threshold of 0.8 on $P\left(C_{p,l}(k) > (2p+1)/(2l+1) \mid \vec{y}\right)$ typically yields an FDR of 5% or less for the experimental data.

We have considered two null distributions in this work, both of which preserve the sequencing coverage within the identified regions as the experimental dataset. The first is a random null distribution, where simulated sequencing reads are assumed to be uniformly distributed within each region. The second is a MNase digestion-aware null, where simulated sequencing reads are assumed to be uniformly distributed subject to the observed distribution of the dinucleotide ends. These two null distributions are applied to identify TBB positions in two scenarios. The MNase digestion-aware null is used to identify experimental TBB positions that are statistically significant over the sequence bias of MNase digestion. The random null is used to set the threshold for determining possible nucleosome positions that result from MNase digestion bias over a uniform background. In all cases, the false discovery rate is controlled to be less than 5%, allowing a fair comparison between different data sets.

## In silico simulation of MNase digestion

For each region of the genome, we tabulated the full table of dinucleotide counts from all aligned paired-end reads. For each pair of cut dinucleotides, we enumerated all potential paired-end reads with matching cut dinucleotides with centers falling in the given region. We then sampled uniformly from this set of potential reads with replacement to match the observed number of reads with the given cut dinucleotides. This yields a sample of reads exactly matching the observed cut dinucleotide distribution with fragment centers random within each region conditional on cut dinucleotides. These simulated controls were then passed through the same pipeline as the observed reads and used to set thresholds based on the stated FDR-controlling procedure.

## Comparison between datasets and replicates

All comparisons are based on matched distances, as in (*Brogaard et al., 2012*). For example, when we compared our identified consensus centers of nucleosomes to those identified from a previous study (*Jiang and Pugh, 2009a*), we calculated the distance between every center in our dataset to its nearest center in the published study and summarized them into distance probability (*Figure 2B–E*) and cumulative (*Figure 2—figure supplement 3–6*) plots. Similarly, when we evaluated the performance of the TBB method on an in silico MNase-seq dataset, we computed the distance between every identified TBB position from the in silico dataset and its nearest simulated nucleosome position. When we evaluated reproducibility between replicates, we considered the set of all best-match distances obtained by matching each replicate against the other to ensure symmetry.

## Random nucleosome positions and consensus centers

We used two methods to generate random nucleosome positions and consensus centers as controls to estimate the detection accuracy of the TBB approach. In the first method, we randomly generated the genomic coordinates of nucleosome positions on each chromosome to match the number of experimentally detected TBB nucleosome positions or consensus centers. In the second method, we took into account the spacing features between TBB nucleosome positions or consensus centers. In this way, the randomly generated genomic coordinates and the experimental determined data have the same distribution of spacing between adjacent positions. The random nucleosome positions or consensus centers maps generated by these two methods yielded similar results and only the result of the second method is shown in the plots (Gray trace, *Figure 2—figure supplements 3–5*). The median distance between the random nucleosome positions and the chemical positions is 18 bp, and is 36 bp for the spacing between the random nucleosome positions and the TBB positions determined here. The median distance between the random consensus centers and either the reference centers or the consensus centers determined here is 49 bp in both cases. The median distance between the random nucleosome positions and the chemical positions is much smaller than the rest of the comparisons because the number of chemical positions was three times larger than the number of TBB nucleosome positions and 5 times larger than the number of consensus centers determined in this study.

## In silico validation of detection accuracy

The true positions of nucleosomes in vivo are generally not available with the current experimental and computational approaches. We thus performed a set of in silico experiments to evaluate the performance of the TBB method in a setting where ground truth of nucleosome positions is available. We first simulated the true positions of nucleosomes: we generated the primary nucleosome positions to represent the most frequent (strongest) positions among a set of overlapping nucleosomes, and then added alternative positions around the primary positions to account for the other overlapping nucleosomes (*Figure 3A*). In each set of in silico experiments, we systematically varied the occupancy (coverage), spacing (offset), and relative strength of primary and alternative nucleosome positions (effective magnitude) according to a factorial design that spans the 5th to 95th percentiles of the corresponding properties observed in our yeast experiments (*Figure 3A*; *Supplementary file 2*). At each of the simulated nucleosome positions, we randomly generated sequencing reads based on the digestion variability template estimated from T1, and constructed 10 artificial chromosomes to represent the in silico MNase-seq data sets. The occupancy of sequencing read midpoints in these simulated data sets resembles that determined from our biological samples. We then applied the TBB approach to identify nucleosome positions in these in silico data sets and compared them with the simulated nucleosome positions (both simulated primary and alternative positions). We found that the TBB approach can reliably identify primary nucleosome positions (50% and 85% of the primary positions within 2 bp and 4 bp, respectively) across all settings. Detection of the alternative positions is similarly reliable (50% and 75% within ~3 bp and ~7 bp, respectively) if the alternative positions are populated at least 1/3 as frequently as the nearest primary positions (effective magnitude smaller than 0.6) (*Supplementary file 1*). Detailed methods and discussion about the in silico validation can be found in Extended Experimental Procedures.

## Procedures for in silico estimation of TBB performance

To estimate the precision of the TBB approach in identifying nucleosome positions, we simulated nucleosome positions on a set of artificial chromosomes, generated in silico MNase-seq sequencing read midpoint datasets based on the experimental digestion variation, estimated the in silico TBB nucleosome positions with these in silico datasets, and compared them to the original simulated nucleosome positions. The differences between these identified TBB in silico positions and the original simulated positions reflect the precision of the TBB approach.

To mimic the organization of nucleosomes in the genome, we simulated nucleosome positions based on observed in vivo organization around genes and constructed simulated artificial chromosomes with units of genes. Each artificial chromosome contains 1100 genes, and each gene was 3501 bp in length, consisting of a 1000 bp promoter region before its transcription start site (TSS) and 2500 bp following the TSS. The in vivo organization of nucleosomes around genes was

determined from the identified consensus centers from the T1 experiment and averaged across all ORFs. As traditionally annotated, the nucleosomes after the TSS were numbered incrementally from +1, and the nucleosomes before TSS were numbered decrementally from −1. The average positions of these nucleosomes relative to TSSs were used in the construction the simulated nucleosome positions. Meanwhile, the number of sequencing reads within each consensus position was used to simulate the in silico MNase-seq datasets.

To test the ability of the proposed model to identify overlapping nucleosomes, we built overlapping nucleosome positions into our simulation. We first generated nucleosome positions downstream of the TSS (corresponding to the positions of the +1, +2, +3, . . . nucleosomes) and upstream of the TSS (corresponding to the positions of the −1, −2, −3, . . . nucleosomes) to represent the most frequent (strongest) positions among a set of overlapping nucleosomes (termed 'primary positions'). Then we added positions around the primary positions (termed 'alternative positions') to represent overlapping nucleosomes. In the simulation, we varied the relationships between the primary positions and the alternative positions to explore the performance of the TBB model. For simplicity, we assumed the alternative positions are symmetric to the primary positions.

We designed a simulation with three factors, varied at the gene level: coverage (the expected number of reads per gene), the spacing between primary nucleosome positions and alternative positions (which we refer to as offset), and the relative magnitudes of primary and alternative positions (which we refer to as effective magnitude and is defined as the percentage of reads attributed to the primary positions) (*Figure 3A*). Coverage had 10 levels, spanning the 5th to 95th percentile observed gene-level coverages in increments of 10%. Alternative position spacing had 10 levels, spanning from 0 bp (no alternative positions) to 45 bp in increments of 5 bp. We tested 11 levels for the relative magnitude between alternative positions and primary positions, spanning from 0 (no alternative positions) to 1 (alternative positions of the same magnitude as primary positions) in increments of 0.1, where the effective magnitude ranged from 1 to 1/3. We used a full factorial design on these three factors, yielding 1100 distinct relationships between the primary and alternative positions for each of 10 simulated chromosomes.

To generate our in silico MNase-seq dataset, we followed a modified version of the generative process described above. For each gene, we first drew coefficients for its subset of $\vec{\beta}$ from an upper-truncated log-normal distributed with parameters estimated from those regions in T1 with similar coverage. These corresponded to 'background' positions and introduce a realistic level of variation into the simulations; biologically, such background could originate from a combination of low-occupancy nucleosome positions and naked DNA obtained during the MNase-seq process.

Then, we set the entries of $\vec{\beta}$ corresponding to the gene's primary and alternative positions deterministically. The sum of the coefficients for these positions was fixed to the total occupancy of the gene minus the sum of the background positions. The relative magnitudes were determined by the design described above, with two alternative positions placed symmetrically around each primary position at the designated spacings. Thus, for a given level of coverage, the expected number of reads within each cluster was fixed, but its distribution across primary and alternative positions varies.

We convolved these $\vec{\beta}$ vectors with the template estimated from the experimental data to obtain vectors of expected read counts $\vec{\lambda}$. Finally, we generated $\vec{y} \sim iid\ Poisson(\vec{\lambda})$ to obtain simulated read counts. This entire procedure was repeated for each replicate, yielding 10 artificial chromosomes of length 3,851,100 bp each. The simulated read midpoint occupancy was similar to the midpoint occupancy observed in vivo around (*Figure 3B*).

Based on our in silico results, the TBB method appears extremely accurate for calling primary nucleosome positions. It can estimate 50% of such positions within 2–3 bp and 95% within 4–5 bp for all simulated conditions, as shown in *Supplementary file 1* Its performance remains strong for the estimation of alternative positions. As the data in *Supplementary file 1* hows, more than 50% of the simulated alternative positions were mapped within 2 bp when the effective magnitude is less than 0.71 (alternative positions populated at least as much as 20% of their corresponding primary positions). When the effective magnitude reaches 0.56 or less (populated as much as 40% of their corresponding primary positions), we mapped over 50% of alternative positions within a single base pair. With the spacing between the alternative and primary positions ranging from 5–45 bp, the

median error for estimating alternative positions is no more than 2 bp. We observed stronger dependence of the TBB method's performance on the spacing between alternative and primary positions: we generally attained higher reliability for the larger offsets, with over 85% of alternative positions estimated within 8 bp when the offset is 10 bp, and within 6 bp when the offset is 40 bp (http://www.github.com/airoldilab/cplate).

## Software availability

All sequencing data are deposited in the NCBI SRA database under accession number SRP023122. All software for the template-based Bayesian model and in silico MNase-seq experiments used in this paper are available at http://www.github.com/airoldilab/cplate.

## Acknowledgements

We thank C Daly, G Marnellos and J Zhang for help with Illumina sequencing; G Basse for help with human nucleosome analysis; BF Pugh and HS Rhee for kindly sharing their +1 nucleosome data; Airoldi lab and O'Shea lab members for discussion and commentary; and W Moebius, O Rando and A Regev for critical reading of the manuscript. This work was supported by the Howard Hughes Medical Institute (EKO), NIH NIGMS grant R01 GM-096193 (EMA) and Alfred P Sloan Research Fellowship (EMA).

## Additional information

### Competing interests

EKO: President at the Howard Hughes Medical Institute, one of the three founding funders of eLife. The other authors declare that no competing interests exist.

### Funding

| Funder | Grant reference number | Author |
| --- | --- | --- |
| Howard Hughes Medical Institute | | Xu Zhou<br>Erin K O'Shea |
| National Institute of General Medical Sciences | R01 GM-096193 | Alexander W Blocker<br>Edoardo M Airoldi |
| Alfred P. Sloan Foundation | | Alexander W Blocker<br>Edoardo M Airoldi |
| Jane Coffin Childs Memorial Fund for Medical Research | | Xu Zhou |

The funders had no role in study design, data collection and interpretation, or the decision to submit the work for publication.

### Author contributions

XZ, AWB, Conception and design, Acquisition of data, Analysis and interpretation of data, Drafting or revising the article; EMA, EKO, Conception and design, Analysis and interpretation of data, Drafting or revising the article

### Author ORCIDs

Xu Zhou, http://orcid.org/0000-0002-1692-6823
Erin K O'Shea, http://orcid.org/0000-0002-2649-1018

## Additional files

### Supplementary files

• Supplementary file 1. A compressed file containing the TBB nucleosome positions and the TBB consensus centers of nucleosomes for yeast data sets 'T1', 'T2', and human chromosome 12, position 38,000,000–48,000,000.

• Supplementary file 2. Cumulative distribution of the distance between in silico TBB positions and matched primary positions (2A) or matched alternative positions (2B) from in silico experiments.

• Supplementary file 3. A table comparison of published methods for determining nucleosome positions.

### Major datasets

The following dataset was generated:

| Author(s) | Year | Dataset title | Dataset URL | Database, license, and accessibility information |
|---|---|---|---|---|
| Xu Zhou, Erin O'Shea | 2013 | A template-based Bayesian method for identifying nucleosome positions at base-pair resolution | http://www.ncbi.nlm.nih.gov/sra/SRP023122/ | Publicly available at the NCBI Short Read Archive (accession no: SRP023122) |

The following previously published datasets were used:

| Author(s) | Year | Dataset title | Dataset URL | Database, license, and accessibility information |
|---|---|---|---|---|
| Brogaard KR, Xi L, Wang J, Widom J | 2012 | A map of nucleosome positions in yeast at base-pair resolution | http://www.ncbi.nlm.nih.gov/geo/query/acc.cgi?acc=GSE36063 | Publicly available at the NCBI Gene Expression Omnibus (accession no: GSE36063) |
| Rhee HS, Pugh BF | 2012 | Genome-wide structure and organization of eukaryotic pre-initiation complexes | http://trace.ncbi.nlm.nih.gov/Traces/sra/?study=SRP010134 | Publicly available at the NCBI Sequence Read Archive (accession no: SRA046523) |

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
