## [Decision Letter]

Thank you for submitting your article "Nucleosome positions and alternative chromatin states revealed by computation approach leveraging digestion variability" for consideration by *eLife*. Your article has been favorably evaluated by Naama Barkai as the Senior Editor, Asifa Akhtar as Reviewing Editor, and two reviewers, including Juan Vaquerizas (Reviewer #2).

The reviewers have discussed the reviews with one another and the Reviewing Editor has drafted this decision to help you prepare a revised submission.

In this study, Zhou et al. introduce a computational method to determine nucleosome positioning from MNase-seq data at a base-pair resolution. Due to digestion variability, it has been a problem comparing nucleosome positioning maps between samples from different MNase-seq experiments. It is also impossible to separate biological variations in nucleosome positioning from technical variations in MNase digestion. The computational method proposed by the authors looks quite promising to solve these problems.

Overall the study is interesting and the method will be of use to the genomics community, since it is currently not straightforward to determine nucleosome positioning with high accuracy.

1) There are currently a number of other methods to determine nucleosome positioning. The manuscript will greatly benefit from performing a comparison with those, so the readers can easily assess the performance of the method.

2) The method describing how to use the software to generate the digestion variability template is too simplified, and the GitHub site is also poorly organized with no documentations, making it very hard to apply the authors' method to other MNase-seq experiments. It limits the usefulness of the authors' approach for the wide audience of *eLife*. This aspect should be improved upon revision.

3) The authors propose there are alternative positioning of nucleosomes near TSSs. Why are these alternative positioning nucleosomes not apparent in the chemical approach (Brogaard et al. 2012)? What will be the overlap between author's approach and the Brogaard approach if you only consider the alternative positioned nucleosomes?

4) The authors suggest that distal nucleosomes might work by providing a boundary for the assembly of the PIC (subsection “Alternative chromatin states and transcription initiation”, third paragraph). However, the authors do not provide enough evidence to be able to make this statement since the distal positioning of the nucleosomes might just reflect the binding of the PIC, and therefore might just be a consequence of the transcriptional process, rather than the provision of a boundary.

5) The authors report a link between the presence of uniquely positioned nucleosomes in highly expressed genes. It would be interesting to stratify genes according to gene expression (for examples using quintiles) and to evaluate whether there are specific enrichments of unique vs alternative nucleosomes across the stratified dataset. This would allow the authors to discern whether the unique vs. alternative nucleosome observation could be explained by the overall level of gene expression or whether the two groups have functional implications. This is important in the context of the subsequent sequence analysis since the authors argue that unique nucleosomes have specific sequence features, which suggest functional implications.

6) In order to assess the functional implications, the manuscript will improve considerably if the unique vs. alternative classification analysis and subsequent gene expression and sequence analyses would be repeated using the human dataset in Figure 4.

---

## [Author Response]

*1) There are currently a number of other methods to determine nucleosome positioning. The manuscript will greatly benefit from performing a comparison with those, so the readers can easily assess the performance of the method.*

We thank the reviewers for this suggestion. The method proposed in this manuscript aims at estimating the position of individual nucleosome configurations defined over a few base pairs, rather than the peak positions of averaged nucleosome occupancy over hundreds of base pairs as with most of the other methods. While it is difficult to make a fair quantitative comparison with other methods, we provide a qualitative comparison below for assessing the performance of the available methods. This information is also summarized in the supplementary material for the readers ([Supplementary-material SD5-data]).

As we described in the manuscript, the proposed method utilizes a digestion template-based Bayesian approach to estimate the probability of observing nucleosomes at base-pair resolution from MNase-seq data. We reported both the average centers of bulk nucleosomes as well the positions of individual nucleosomes. Parameters used in the model are inferred directly from experimental datasets. In comparison, the Parzen window-based methods, as used by Albert et al. (2007), Valouev et al. (2008), Tsankov et al., (2010) and many others, estimate the peak of smoothed nucleosome occupancy along relatively broad nucleosome-containing regions. The number and locations of nucleosomes are highly dependent on smoothing window widths that are defined by users. Weiner et al. (2010) proposed a template-based filtering approach to deal with variable nucleosome sizes and fuzzy digestion ends. However, it cannot provide fine-grained mapping of overlapping nucleosome positions. The NOrMAL method developed by Polishko et al. (2012) is a recent method with improved identification of non-overlapping nucleosome positions compared with previous methods, but still depends on user defined parameters and cannot map positions of individual nucleosomes. In a separate simulation analysis, we described a direct comparison between the performance of NOrMAL and the performance of the template-based Bayesian method (Blocker and Airoldi 2016, Section 4). The TBB method outperforms NOrMAL in all the settings we have considered. We conclude that our work provides the best resolution and most accurate method to identify positions of individual nucleosomes from MNase-seq data.

In addition to methods developed for MNase-seq data, alternative approaches utilizing chemical cleavage (Bogaard et al., 2012), ATAC-Seq (Schep et al., 2015) and DNase- seq (Zhong et al., 2016) provide complementary approaches to determining nucleosome position genome-wide. The nucleosome positions in *S. cerevisiae* determined by chemical cleavage have been regarded as the gold-standard in the field. We provided direct comparison with the chemical cleavage method in the manuscript, in Figure 2 and Figure 2—figure supplement 4.

In all, each computational and experimental method has their advantages and limitations. To help the reader easily assess the performance of different methods for determining nucleosome positions, we summarize the above description and references in [Supplementary-material SD5-data]. We added the following text to introduce the comparison.

“Several methods have been developed to determine the genome-wide locations of nucleosomes, each of which addresses different experimental and computational issues (Albert et al., 2007; Brogaard et al., 2012; Polishko et al., 2012; Schep et al., 2015; Tirosh, 2012; Tsankov et al., 2010; Valouev et al., 2011; Weiner et al., 2010; Zhong et al., 2016). We briefly summarize the approaches, performance and limitations of these methods for reference ([Supplementary-material SD5-data]).”

*2) The method describing how to use the software to generate the digestion variability template is too simplified, and the GitHub site is also poorly organized with no documentations, making it very hard to apply the authors' method to other MNase-seq experiments. It limits the usefulness of the authors' approach for the wide audience of eLife. This aspect should be improved upon revision.*

We have updated the documentation of the software on GitHub (http://www.github.com/airoldilab/cplate) to guide users on how to apply the proposed methods to other MNase-seq experiments. The updated GitHub repository includes documentation on software requirements, as well as step-by-step instructions on how to install and replicate the template-based analysis of MNase-seq data. The GitHub repository also includes two examples for demonstration: one using simulated MNase- seq data including the software code to run the simulation, and another one using human MNase-seq data published in Gaffney et al. (2012). The examples can be found at https://github.com/airoldilab/cplate/tree/master/examples/human.

*3) The authors propose there are alternative positioning of nucleosomes near TSSs. Why are these alternative positioning nucleosomes not apparent in the chemical approach (Brogaard et al. 2012)? What will be the overlap between author's approach and the Brogaard approach if you only consider the alternative positioned nucleosomes?*

We analyzed the same genomic regions shown in Figure 5 with the chemical nucleosome map from Brogaard et al. 2012, and observed several alternative nucleosome positions present around the TSS. On average, 5.8 chemical positions were found within each genomic region examined in Figure 5. Next, we examined the overlap between our unique and alternative nucleosome positions (both the distal and proximal positions) near TSS and the chemical nucleosome positions, by calculating the minimum distance between a TBB nucleosome position and a chemical position. As shown in Figure 8, half of the unique positions and half of alternative positions are within 3 bp and 4 bp of a Brogaard chemical position, respectively. 75% of the unique positions are within 11 bp and 75% of the alternative positions are within 13 bp. These data suggest that the overlap between the TBB nucleosome positions and the chemical nucleosome positions are similar between regions near gene TSSs and in other regions across the whole genome (Figure 2—figure supplement 4).

Author response image 1.Cumulative distribution of the nearest distance analysis for TBB positions around gene TSSs.The curves show the cumulative distribution of the nearest distance between the TBB nucleosome positions at gene TSSs analyzed for Figure 5 and the nucleosome positions mapped by the chemical approach. Uniquely positioned nucleosomes and alternatively positioned nucleosomes (both distal and proximal nucleosomes) are analyzed separately.**DOI:**
http://dx.doi.org/10.7554/eLife.16970.029

Overall, both the unique and alternative nucleosome positions identified in our work near TSSs agreed well with the nucleosome positions identified by the chemical cleavage approach. Although it is comprehensive, the chemical nucleosome map makes it difficult to distinguish if one or a few nucleosome positions are the dominant positions. In comparison, the TBB approach reveals alternative nucleosome positions that are dominant within a population of cells, and exposes the pattern of alternative nucleosomes around gene TSSs.

*4) The authors suggest that distal nucleosomes might work by providing a boundary for the assembly of the PIC (subsection “Alternative chromatin states and transcription initiation”, third paragraph). However, the authors do not provide enough evidence to be able to make this statement since the distal positioning of the nucleosomes might just reflect the binding of the PIC, and therefore might just be a consequence of the transcriptional process, rather than the provision of a boundary.*

We agree with the reviewer’s comment and have changed the original text “suggesting that the 1 nucleosome in the distal position provides a boundary for the assembly of the PIC” to “suggesting that the 1 nucleosome in the distal position marks a boundary for the assembly of the PIC. This boundary may be provided by the positioned nucleosomes or by binding of one or more components of the transcription complex”.

*5) The authors report a link between the presence of uniquely positioned nucleosomes in highly expressed genes. It would be interesting to stratify genes according to gene expression (for examples using quintiles) and to evaluate whether there are specific enrichments of unique vs. alternative nucleosomes across the stratified dataset. This would allow the authors to discern whether the unique vs. alternative nucleosome observation could be explained by the overall level of gene expression or whether the two groups have functional implications. This is important in the context of the subsequent sequence analysis since the authors argue that unique nucleosomes have specific sequence features, which suggest functional implications.*

We have performed the analysis suggested by the reviewers to further examine the relationship between alternative positions at TSSs and gene expression level. Following our original analysis, we stratified genes shown in Figure 6 (the same as in Figure 5) based on their expression level, and evaluated whether there is any specific enrichment of unique vs alternative nucleosomes across stratified data. We obtained the expression levels of 3368 genes from Newman et al. 2006 when yeast cells are grown in synthetic complete medium, ranked them from highest to lowest, and evaluated the enrichment in unique nucleosomes in either 4, 5, 8, or 10 stratified segments. The P-values are calculated with Fisher’s exact test for each segment and are plotted below (Figure 9). Similar to our description in the text, uniquely positioned nucleosomes are enriched among the highest expressed genes.

Author response image 2.Significance of the enrichment in uniquely positioned nucleosomes among genes stratified by expression level determined in Newman et al.**DOI:**
http://dx.doi.org/10.7554/eLife.16970.030

Since the Newman et al. dataset only covers ~ 75% of genes examined here, we extended our analysis to an RNA-seq dataset of yeast cells cultured at similar conditions (Zid at al., 2014), where expression of all of the genes can be analyzed. After performing the same analysis described above, we observed similar results as for the Newman et al. dataset (Figure 10), suggesting that uniquely positioned nucleosomes are only enriched in the most highly expressed genes. We fail to observe significant enrichment in alternatively positioned nucleosomes in any stratified data we examined so far.

Author response image 3.Significance of the enrichment in uniquely positioned nucleosomes among genes stratified by expression level determined in Zid et al.**DOI:**
http://dx.doi.org/10.7554/eLife.16970.031

Overall, we concluded that the unique vs. alternative nucleosome positions at TSSs cannot be fully explained by the overall level of gene expression. We echo the reviewer’s opinion that there may be other functional implications of unique vs alternative nucleosomes. Our work provides the tool to finely dissect chromatin configurations and investigate the relationship between these configurations and gene regulation.

*6) In order to assess the functional implications, the manuscript will improve considerably if the unique vs. alternative classification analysis and subsequent gene expression and sequence analyses would be repeated using the human dataset in Figure 4.*

We thank the reviewer for this suggestion. Human nucleosome datasets from Gaffney et al. 2012 were constructed from pooling paired-end MNase-seq data from seven human lymphoid cell lines. Each cell line may differ in their genetic backgrounds, gene expression, chromatin state, etc. These differences likely all contribute to positioning of nucleosomes. We cannot conclude whether overlapping nucleosome positions identified from these datasets originate from differences between cell lines or mechanisms related to gene regulation. It is also difficult to properly correlate promoter nucleosome configurations with gene expression, transcription machinery binding etc. We thus think the nature of the datasets renders it unsuitable for the comprehensive characterization suggested.